# Auto-Rubric: Learning to Extract Generalizable Criteria for Reward Modeling

## Abstract

Reward models are essential for aligning Large Language Models (LLMs) with human values, yet their development is hampered by costly preference datasets and poor interpretability. While recent rubric-based approaches offer transparency, they often lack systematic quality control and optimization, creating a tension between scalability and reliability. We address these limitations with a novel, training-free framework built on a key assumption: *evaluation rubrics underlying human preferences exhibit strong generalization ability across diverse queries*, a property that enables remarkable data efficiency. Our two-stage approach first infers high-quality, query-specific rubrics using a validation-guided **Propose-Evaluate-Revise** pipeline. Second, it generalizes these granular rubrics into a compact, non-redundant core set by maximizing an **information-theoretic coding rate**. The final output is an interpretable, hierarchical "Theme-Tips" rubric set. Extensive experiments demonstrate the framework's exceptional data efficiency and performance. Critically, using just 70 preference pairs (1.5% of the source data), our method empowers smaller models like Qwen3-8B to outperform specialized, fully-trained counterparts while providing full interpretability of the evaluation process. This work pioneers a scalable, interpretable, and data-efficient path for reward modeling. Related code and data are available at https://anonymous.4open.science/r/Auto-Rubric-9219/.

## 1 Introduction

Reinforcement Learning from Human Feedback (RLHF) is a powerful paradigm for aligning Large Language Models (LLMs) with human values (Ouyang et al., 2022). As shown in Figure 1, the core of RLHF is a reward model (RM) trained on vast datasets of human preferences to serve as a proxy for human judgment (Gao et al., 2023; Guo et al., 2025). However, this approach is fundamentally limited by the prohibitive cost of data acquisition and the "black-box" nature of the reward models (Liu et al., 2025a). This lack of interpretability not only hinders our ability to diagnose failures but also elevates the risk of "reward hacking" (DeepSeek-AI et al., 2025), where models exploit the proxy in unintended ways.

To address these shortcomings, rubric-based evaluation using explicit criteria has gained traction as a more transparent alternative. The rubric is a set of explicit human-readable criteria, such as factual accuracy and well-organized content, which can be effectively integrated as part of the prompt for the "LLM-as-a-Judge" paradigm. Early approaches relied on expert-defined rubrics (Hashemi et al., 2024) or large-scale crowd annotations (Bai et al., 2022), but their limited scalability prompted a shift towards automated rubric generation (Wang & Xiong, 2025; Gupta et al., 2025). These methods often produce rubrics that suffer from noise, redundancy, and misalignment with human preferences due to the lack of a verification mechanism. Consequently, a fundamental tension arises between scalability and fidelity, which poses the primary bottleneck for the broader adoption of rubric-based evaluation.

To resolve this tension, we propose a new framework for the automated generation and refinement of high-quality evaluation rubrics with a small corpus of preference data. Our work is built on a key assumption: *evaluation rubrics underlying human preferences exhibit strong generalization ability across diverse queries*. This assumption is grounded in the theory of **cognitive universality**: research in judgment and decision-making has shown that humans evaluate information using a

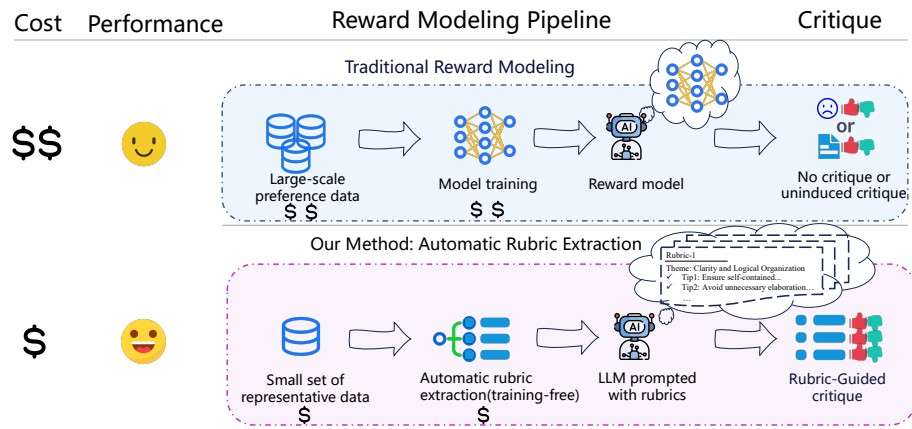

Figure 1: Comparison between traditional reward model training and our automatic rubric extraction method. Traditional methods require large-scale preference data and costly training to produce reward models. In contrast, our method uses a small set of representative data to automatically extract interpretable "Theme-Tips" rubrics for reward modeling, achieving low cost and high interpretability.

stable set of high-level heuristics (e.g., clarity, factual accuracy, and logical coherence) that remain consistent across domains and contexts (Kahneman, 2011). We empirically validate this assumption in Section G, where rubrics extracted from conversational data successfully transfer to specialized domains (Math, Code, and Safety), providing concrete evidence for our theoretical premise. Our goal is not to learn an opaque reward function but to explicitly infer the underlying criteria—the rubric—that govern human choices. This represents a fundamental shift from typical **reward model learning** to **rubric learning**, a contrast visually summarized in Figure 1.

To operationalize this new paradigm, our approach operates through two stages: **Query-Specific Rubric Generation** and **Query-Agnostic Rubric Aggregation**. First, Query-Specific Rubric Generation employs an iterative **Propose-Evaluate-Revise** loop that treats rubric generation as a constrained optimization problem, ensuring each rubric is validated for its discriminative power. Second, Query-Agnostic Rubric Aggregation uses an **information-theoretic selection** algorithm to distill the large pool of validated, granular rubrics into a compact, hierarchically structured rubric we term the **"Theme-Tips" hierarchy**. This structure addresses the trade-off between generality and specificity:

- **Theme:** Represents a high-level, universal evaluation dimension applicable across diverse queries.

- **Tips:** Consists of specific, actionable constraints that provide granular guidance for the model.

Our primary contributions are as follows:

- **A data-efficient, training-free framework for automated rubric extraction.** Our two-stage Propose-Evaluate-Revise and information-theoretic selection mechanism achieves state-of-the-art performance using only a fraction of typical preference data.

- **Open-source rubric datasets.** We release public datasets of query-agnostic rubrics inferred from preference data to facilitate research into interpretable alignment.

- **A quantitative framework for analyzing individual rubric utility.** We introduce a quantitative method to dissect rubric utility via Coverage, Precision, and Contribution metrics, offering deeper insight into the evaluation process.

- **State-of-the-art performance on reward modeling benchmarks.** Our method consistently improves base LLMs across four benchmarks. Notably, our rubric-enhanced Qwen3-8B (80.91% on RewardBench2) surpasses the specialized, fully-trained Skywork-Reward-V2-Qwen3-8B (78.20%), demonstrating that our training-free method empowers smaller models to match or exceed resource-intensive training-based approaches.

## 2  RELATED WORK

**LLM-as-a-Judge for Automated Evaluation.**  The "LLM-as-a-Judge" paradigm has emerged as a transformative framework for automated evaluation, leveraging the reasoning capabilities of LLMs to assess model outputs without extensive human intervention. Seminal works such as MT-Bench (Zheng et al., 2023) and Arena-Hard (Li et al., 2024b) have demonstrated that LLM judges can achieve strong correlation with human preferences while offering unprecedented scalability. Fundamentally, this approach relies on prompt engineering, where zero-shot and few-shot strategies guide models to perform complex evaluation tasks. However, relying solely on unstructured prompting often introduces significant inconsistency and contextual biases (Wang et al., 2024b). These limitations necessitate the development of more structured and robust evaluation frameworks (Chen et al., 2024; Li et al., 2023).

**Rubric-Based Reward Modeling.**  To improve reliability, recent work incorporates explicit criteria (rubrics) or explicit reasoning processes into reward modeling. Critique-generating reward models such as DeepSeek-GRM (Liu et al., 2025c), Auto-J (Li et al., 2023), and TIGERScore (Jiang et al., 2024) are trained to produce natural language critiques alongside scores, but require computationally intensive training to embed evaluation standards directly into model parameters. In contrast, we keep evaluation criteria as portable prompt-context rubrics rather than implicit weights, avoiding the high cost of specialized training while maintaining transparency. Another line of work, including HealthBench (Arora et al., 2025), RewardAnything (Yu et al., 2025), LMUnit (Saad-Falcon et al., 2024), and Rule-based Rewards (Mu et al., 2024), relies on static, expert-curated rubrics or hand-written rules, leading to a "cold-start" bottleneck and poor scalability to new domains. Our approach instead infers high-quality rubrics directly from raw preference data, effectively bridging the gap between data and structured evaluation without manual intervention.

**Inverse Constitutional AI.**  Inverse Constitutional AI (ICAI) (Henneking & Beger, 2025; Findeis et al., 2025; An, 2025) aims to reverse-engineer the explicit rubrics (or "constitution") that drive human preferences by analyzing feedback patterns and distilling latent alignment rules into interpretable formats. However, existing approaches typically assume that rubrics derived from feedback are inherently correct, without mechanisms for active verification or correction, and they often operate at the instance level, producing a fragmented collection of query-specific rules that lack a coherent global organization (Henneking & Beger, 2025). In practice, ICAI systems also face challenges with redundancy and conflicts between extracted rules, and most work provides limited guidance on how to aggregate, prioritize, or prune rubrics for scalable use in downstream evaluation or training.

## 3  METHODOLOGY

**Overview.**  Our framework systematically infers a general-purpose, interpretable set of evaluation rubrics from a small sample of human preferences. The methodology is structured into several stages, beginning at a granular level to maximize data efficiency. We first formulate rubric learning as an alternative to traditional reward modeling. Next, in the initial generation stage, a small seed batch is processed to infer high-fidelity, query-specific rubrics for each preference pair through a validation-centric loop, as illustrated in Figure 2. These granular rubrics are then aggregated into a compact, query-agnostic set using an information-theoretic approach. Finally, we introduce a quantitative framework to analyze the utility and contribution of each rubric in the final set.

### 3.1  FORMULATION

The conventional approach to learning from human preferences is to train a parametric reward model. Given a preference dataset $\mathcal{D} = \{(x_i, y_i^+, y_i^-)\}_{i=1}^N$, the goal is to learn a scalar reward function $r_\theta(x, y)$ that assigns a higher score to the preferred response. This is often optimized using a Bradley-Terry model (Bradley & Terry, 1952), where the probability of a preference is modeled as:

$$P(y_i^+ \succ y_i^- | x_i) = \sigma(r_\theta(x_i, y_i^+) - r_\theta(x_i, y_i^-)). \tag{1}$$

The objective is to find the optimal parameters $\theta$ by maximizing the log-likelihood over the dataset. While effective, this process yields an opaque reward function $r_\theta$—a "black box" that offers limited

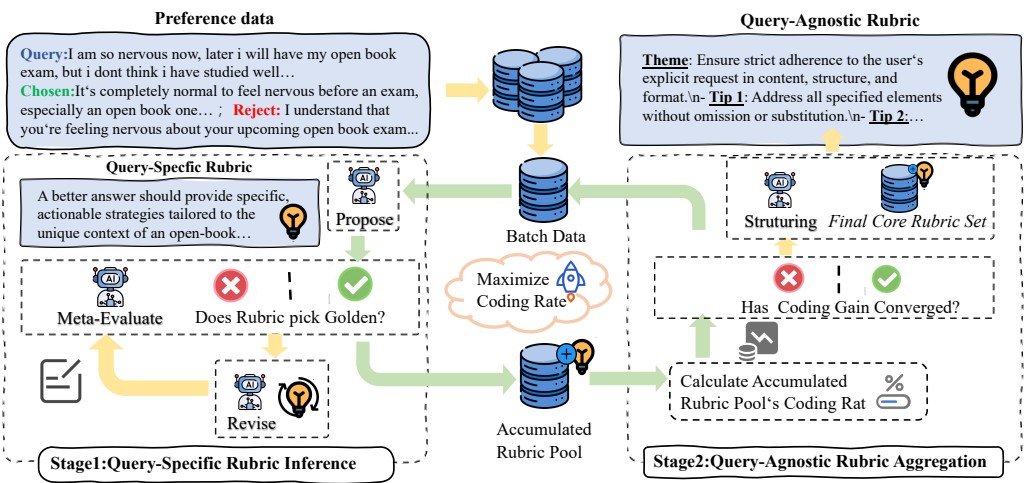

Figure 2: Overview of our two-stage rubric extraction framework. **Stage 1 (Query-Specific Rubric Inference):** An iterative Propose-Evaluate-Revise pipeline generates high-quality, query-specific rubrics from batches of preference data. **Stage 2 (Query-Agnostic Rubric Aggregation):** An information-theoretic selection algorithm distills the accumulated rubric pool into a compact, non-redundant core set by maximizing coding rate. The final output is structured into an interpretable "Theme-Tips" hierarchy for enhanced usability and generalization.

insight into why one response is preferred over another. This lack of interpretability hinders failure diagnosis and trust.

To overcome these challenges, our work attempts a paradigm shift from **Reward Model Learning** to **Rubric Learning**. Instead of optimizing the parameters $\theta$ of an inscrutable function, our objective is to directly infer the explicit, human-readable rubric set $R$ that best explains the preferences in $\mathcal{D}$. Our optimization problem remains:

$$R_{\text{task}}^* = \arg\max_R \sum_{i=1}^{N} \mathbb{I}[\text{eval}_R(x_i, y_i^+, y_i^-) = \text{correct}]. \tag{2}$$

However, the evaluation function, $\text{eval}_R(\cdot)$, is no longer a parametric model but a prompt-guided judgment procedure conditioned on the natural language rubric in $R$. In practice, this evaluation function is implemented by prompting a large language model with the query, candidate responses, and the rubric set $R$, tasking it to make a preference judgment. Directly solving for $R_{\text{task}}$ in Equation 2 is intractable, as it requires searching an extremely large and unstructured space of natural language rules. To make this problem tractable, we decompose it into two complementary sub-problems: first generating validated candidate rubrics for individual samples, then selecting a diverse, non-redundant subset that generalizes across queries.

## 3.2 QUERY-SPECIFIC RUBRIC GENERATION

Instead of requiring a large-scale dataset, our framework begins at a granular level by processing a small seed batch to infer high-quality rubrics for each individual preference pair $(x_i, y_i^+, y_i^-)$. The core of this process is an iterative **Propose-Evaluate-Revise** loop, which emphasizes verification to ensure rubric quality.

Formally, the process for a single preference pair begins with a proposal model, $\mathcal{M}_{\text{propose}}$, proposing an initial rubric set:

$$R_i^{(0)} \leftarrow \mathcal{M}_{\text{propose}}(x_i, y_i^+, y_i^-). \tag{3}$$

At each iteration $t$, an evaluation model, $\mathcal{M}_{\text{evaluate}}$, verifies the current rubric set $R_i^{(t)}$ by making a judgment:

$$y_{\text{pred}}^{(t)} \leftarrow \mathcal{M}_{\text{evaluate}}(x_i, y_i^+, y_i^-, R_i^{(t)}). \tag{4}$$

This verification step is necessary, acting as a quality gate. If the prediction does not match the ground-truth preference ($y_{\text{pred}}^{(t)} \neq y_i^+$), the failed rubric set $R_i^{(t)}$ is used as negative feedback. A revision model, $\mathcal{M}_{\text{revise}}$, then produces an improved set:

$$R_i^{(t+1)} \leftarrow \mathcal{M}_{\text{revise}}(x_i, y_i^+, y_i^-, R_i^{(t)}). \tag{5}$$

This iterative refinement continues until verification succeeds or a maximum number of iterations $E_{\max}$ is reached.

Finally, for each sample $(x_i, y_i^+, y_i^-)$, we generate a query-specific rubric set $R_i^*$ that captures the most relevant evaluation criteria for that specific instance. This process populates a large pool of candidate rubrics, $\mathcal{R}_{\text{pool}} = \bigcup_{i=1}^N R_i^*$.

### 3.3 Query-Agnostic Rubric Aggregation

The query-specific generation stage yields a candidate pool $\mathcal{R}_{\text{pool}}$ of validated rubrics. However, this pool exhibits two critical limitations that prevent its direct use: (1) **Semantic redundancy**: the same underlying criterion (e.g., "clarity") appears in many slightly different phrasings across samples; and (2) **Over-specificity**: many rubrics are too narrowly tailored to their source query to generalize effectively. Therefore, a query-agnostic aggregation stage is essential to distill a minimal yet comprehensive core set of rubrics that enhances generalization and transferability to unseen queries. This is achieved by identifying and consolidating the most essential and recurrent evaluation rubrics from the raw, query-specific pool.

To achieve this, we select a subset that maximizes information gain, ensuring high semantic coverage while minimizing redundancy. **Geometrically, this is equivalent to selecting a set of embedding vectors that span the largest possible volume, a process that naturally penalizes redundant (i.e., near-collinear) vectors.** Our selection criterion, the maximization of the **coding rate** (Yu et al., 2020), directly operationalizes this principle. It is an information-theoretic measure defined on the rubric embeddings $\mathbf{E}_R \in \mathbb{R}^{d \times |R|}$:

$$\mathcal{C}(\mathbf{E}_R, \varepsilon) = \frac{1}{2} \log \det \left( \mathbf{I} + \frac{1}{\varepsilon^2 |R|} \mathbf{E}_R^\top \mathbf{E}_R \right), \tag{6}$$

where $\mathcal{C} \in \mathbb{R}$ and $\varepsilon > 0$ controls the trade-off between compression and fidelity. Maximizing this function is equivalent to maximizing the volume spanned by the rubric embedding vectors, thus promoting diversity. The optimization problem is to find the core set $R_{\text{core}}$:

$$R_{\text{core}}^* = \arg \max_{R \subseteq \mathcal{R}_{\text{pool}}, |R| \leq m} \mathcal{C}(\mathbf{E}_R, \varepsilon), \tag{7}$$

where $m$ is the desired size of the rubric set. Since this problem is NP-hard, we employ a greedy algorithm that iteratively selects the rubric providing the highest marginal information gain. Starting with an empty set $R_0 = \emptyset$, at each step $k$, we add the rubric $r_{k+1}$ such that:

$$r_{k+1} = \arg \max_{r \in \mathcal{R}_{\text{pool}} \setminus R_k} \left[ \mathcal{C}(\mathbf{E}_{R_k \cup \{r\}}, \varepsilon) - \mathcal{C}(\mathbf{E}_{R_k}, \varepsilon) \right]. \tag{8}$$

This process continues until convergence, which is determined by an early-stopping criterion: the marginal gain in coding rate must fall below a minimum threshold ($\tau_{\min}$) for a set number of consecutive iterations ($p_{\text{patience}}$) to ensure the information content of the core set has saturated. Finally, the selected core set $R_{\text{core}}$ is structured into our interpretable **"Theme-Tips" hierarchy**. This structuring is achieved by prompting the backbone LLM to perform semantic clustering and summarization on the selected rubrics. As detailed in the **Rubric Structuring Prompt** (Appendix K, Figure 10), the model synthesizes fragmented rules into these cohesive Theme-Tip pairs.

### 3.4 A Framework for Rubric Analysis

To ensure the final rubric set is not only performant but also robust and well-structured, we introduce a quantitative analysis framework. This framework, a core part of our methodology, allows us to

---

**Algorithm 1** Batch-Iterative Rubric Extraction

---

**Input:** Seed preference dataset $\mathcal{D}_0$.
**Output:** Structured query-agnostic rubric set $R_{\text{task}}$.
1: Initialize an empty core rubric set $R_{\text{core}} \leftarrow \emptyset$.
2: **for** each batch-iteration $t = 1, 2, \ldots, T$ **do**
3:      Sample a mini-batch of preferences $D_{\text{batch}}$ from $\mathcal{D}_0$.
4:      Generate query-specific rubrics $\mathcal{R}_{\text{new}}$ using the **Propose-Evaluate-Revise** loop (Sec. 3.2).
5:      Form a candidate pool: $\mathcal{R}_{\text{pool}} \leftarrow R_{\text{core}} \cup \mathcal{R}_{\text{new}}$.
6:      Update the core set $R_{\text{core}}$ by selecting rubrics from $\mathcal{R}_{\text{pool}}$ (Eq. 7,8).
7: **end for**
8: Structure the final set $R_{\text{core}}$ to yield $R_{\text{task}}$.
9: **return** $R_{\text{task}}$.

---

dissect the utility of each individual rubric within the final set $R_{\text{task}}$. By evaluating each rubric along three key dimensions, as defined in (Eq. 9,10,11), we can validate the effectiveness of our aggregation process and gain deeper insights into the evaluation mechanism.

For each rubric $r_j \in R_{\text{task}}$, we define the following metrics:

- **Coverage:** The proportion of test samples where the rubric provides a discriminative signal. This metric measures the rubric's generality and applicability.

$$\text{Coverage}(r_j) = \frac{1}{|D_{\text{test}}|} \sum_{i \in D_{\text{test}}} \mathbb{I}[\text{eval}_{\{r_j\}}(x_i, y_i^+, y_i^-) \neq \text{tie}]. \tag{9}$$

- **Precision:** The conditional probability that the rubric's judgment aligns with the ground truth, given that it provides a discriminative signal. This measures the rubric's reliability.

$$\text{Precision}(r_j) = P(\text{eval}_{\{r_j\}} \text{ is correct} | \text{eval}_{\{r_j\}} \neq \text{tie}). \tag{10}$$

- **Contribution:** The marginal impact of a rubric on the full set's performance, measured by the drop in overall accuracy upon its removal. This quantifies the rubric's unique value and non-redundancy.

$$\text{Contribution}(r_j) = \text{Acc}(R_{\text{task}}) - \text{Acc}(R_{\text{task}} \setminus \{r_j\}). \tag{11}$$

This analytical framework is crucial for verifying that our method produces a complementary set of rubrics, balancing general, high-coverage rubrics with specialized, high-precision ones.

## 4 EXPERIMENT

In this section, we conduct a series of experiments to validate our framework's core contributions, showing: (1) state-of-the-art performance on standard reward modeling benchmarks; (2) high data efficiency through rapid convergence; and (3) strong downstream gains in policy optimization.

### 4.1 EXPERIMENTAL SETTING

**Datasets.** We extract rubrics from two preference datasets: (1) **HelpSteer3-Preference** (Wang et al., 2025) provides open, human-annotated preferences spanning four domains: General, STEM, Code, and Multilingual. We focus on the General domain for rubric extraction. (2) **UltraFeedback-Binarized** (Cui et al., 2024) contains prompts with model completions scored by GPT-4 on rubrics such as helpfulness and honesty.

**Baselines.** We compare against four classes of baselines: (1) **Base models**: zero-shot evaluation without rubrics. (2) **In-context learning (ICL)**: the same models prompted with $k = 5$ preference examples (Dong et al., 2022). (3) **Training-based reward models**: state-of-the-art systems including ArmoRM, J1, R3, RM-R1, and Skywork-Reward-V2 (Wang et al., 2024a; Whitehouse et al., 2025; Anugraha et al., 2025; Chen et al., 2025; Liu et al., 2025a). (4) **LLM-as-a-Judge configurations**: expert-curated prompts from Arena-Hard and MT-Bench, and the ICAI protocol (Li et al., 2024b; Zheng et al., 2023; Findeis et al., 2025).

**Evaluation Benchmarks.** We evaluate on four standard benchmarks covering diverse domains: RewardBench (Lambert et al., 2024), Rewardbench2 (Malik et al., 2025), RM-Bench (Liu et al., 2025b), JudgeBench (Tan et al., 2025).

**Models.** Our training-free framework uses **Qwen3-32B** (Yang et al., 2025) as the backbone for rubric construction—including proposal, evaluation, revision, and structuring. We then evaluate these rubrics with multiple LLM families (Qwen3, Claude-4-Sonnet, and GPT-4o), and find that rubrics generated by Qwen3-32B exhibit the strongest cross-model transfer (see Appendix D).

**Policy Optimization.** We evaluate the downstream impact of our rubrics by finetuning **Qwen2.5-7B-Instruct** with Direct Preference Optimization (DPO) (Rafailov et al., 2024) on **WildChat** (Zhao et al., 2024) prompts labeled by our Auto-Rubric evaluator.

Detailed experimental settings and implementation details are provided in Appendix C.

## 4.2 MAIN RESULTS

**State-of-the-Art Performance Across Benchmarks.** As shown in Table 1, our framework achieves state-of-the-art performance across all four evaluation benchmarks. When applied to Claude-4-Sonnet, our method achieves top scores of **95.81%** on RewardBench, **87.90%** on RewardBench2, **89.49%** on RM-Bench, and **81.71%** on JudgeBench, outperforming all baseline approaches including training-based reward models and other LLM-as-a-Judge methods. This broad success highlights the robustness and general applicability of our optimized rubric extraction approach.

**Consistent Improvement Across All Models and Methods.** Our framework consistently improves performance across diverse model families and scales. Relative to base models, Qwen3-8B gains +6.54% on RewardBench2, Qwen3-32B gains +6.72%, and Claude-4-Sonnet gains +1.20%. Notably, our rubric-guided Qwen3-8B surpasses the specialized, fully trained Skywork-Reward-V2-Qwen3-8B on RewardBench2 (**80.91%** vs. 78.20%) and RM-Bench (**88.60%** vs. 82.60%), showing that a training-free approach can outperform resource-intensive reward models.

**Robustness Across Rubric Source Datasets.** Our framework generalizes well across data sources. Rubrics extracted from human-annotated HelpSteer3 and AI-labeled UltraFeedback both achieve state-of-the-art results with complementary strengths: for Claude-4-Sonnet, HelpSteer3 rubrics lead on RewardBench (95.81%) and RM-Bench (89.49%), while UltraFeedback rubrics are best on RewardBench2 (87.90%) and JudgeBench (81.71%), indicating that our method captures fundamental preference patterns from both human and AI annotations.

## 4.3 DATA EFFICIENCY AND CONVERGENCE ANALYSIS

Our framework achieves high performance with very little data because it performs **discrete combinatorial optimization**: instead of updating continuous parameters via repeated passes over large datasets, it selects an optimal subset of natural language rubrics from a candidate pool. Two mechanisms drive this efficiency.

**Query-Specific Extraction with Validation.** Each preference pair yields a validated rubric through a small number of refinement cycles (typically 3–5; Figure 5). Unlike gradient-based updates that require thousands of samples to converge, a single well-validated rubric captures a reusable evaluation criterion, and the Propose-Evaluate-Revise loop ensures that every accepted rubric has verified discriminative power.

**Information-Theoretic Deduplication with Early Stopping.** We process samples in mini-batches of size $B = 10$, greedily selecting rubrics that maximize semantic diversity under the coding rate objective (Eq. 6). Once the rubric space is well covered, additional samples contribute only redundant rubrics and the selection process halts automatically. On HelpSteer3, this occurred after 7 batches (**70 samples**, 1.5% of the source data), yielding $k = 5$ Theme-Tips rubrics. Figure 3 summarizes these convergence dynamics: the t-SNE plot (Figure 3a) shows early batches rapidly spanning diverse semantic clusters, while the marginal coding rate gain $\Delta\mathcal{C}$ in Figure 3b turns negative after batch 5 as the complexity penalty $\frac{1}{\varepsilon^2 |R_{core}|}$ outweighs gains from redundant rubrics. This

Table 1: Performance comparison on four key benchmarks (in %). Methods are grouped into **Training-based** approaches (requiring model fine-tuning) and **Training-free** approaches (prompt-based). Our Auto-Rubric is training-free yet outperforms trained reward models.

| Method Type | Model / Variant | RewardBench | RewardBench2 | RM-Bench | JudgeBench |
|---|---|---|---|---|---|
| **Training-based Reward Models** | | | | | |
| | ArmoRM-Llama3-8B-v0.1 | 90.40 | 66.50 | 69.30 | 59.70 |
| | J1-Llama-8B | 85.70 | – | 73.40 | 42.00 |
| | J1-Llama-70B | 93.30 | – | 82.70 | 60.00 |
| | R3-QWEN3-8B-14K | 87.50 | – | 82.10 | – |
| | R3-QWEN3-14B-LORA-4K | 89.30 | – | 84.90 | – |
| | RM-R1-Qwen-Instruct-32B | 92.90 | – | 79.10 | – |
| | RM-R1-DeepSeek-Distill-Qwen-32B | 90.90 | – | 83.90 | – |
| | Skywork-Reward-V2-Qwen3-8B | 93.70 | 78.20 | 82.60 | 73.40 |
| **Training-free Methods** | | | | | |
| *Zero-Shot Base Models* | | | | | |
| | Qwen3-8B | 92.93 | 74.37 | 86.90 | 73.14 |
| | Qwen3-32B | 92.96 | 75.55 | 85.67 | 75.71 |
| | GPT-4o | 88.24 | 72.00 | 72.80 | 68.29 |
| | Claude-4-Sonnet | 94.61 | 86.70 | 85.70 | 78.29 |
| *In-Context Learning (k=5)* | | | | | |
| | Qwen3-8B | 90.18 | 72.57 | 86.83 | 67.71 |
| | Qwen3-32B | 90.82 | 75.24 | 85.91 | 74.00 |
| | GPT-4o | 88.89 | 73.11 | 74.02 | 68.91 |
| | Claude-4-Sonnet | 94.82 | 84.89 | 83.29 | 77.61 |
| *LLM-as-a-Judge: Arena-Hard* | | | | | |
| | Qwen3-8B | 85.63 | 78.93 | 85.88 | 78.57 |
| | Qwen3-32B | 89.95 | 83.22 | 87.09 | 71.43 |
| | GPT-4o | 89.08 | 74.10 | 79.56 | 68.86 |
| | Claude-4-Sonnet | 95.41 | 84.99 | 88.11 | 82.57 |
| *LLM-as-a-Judge: MT-Bench* | | | | | |
| | Qwen3-8B | 92.56 | 73.40 | 84.84 | 77.14 |
| | Qwen3-32B | 91.99 | 74.29 | 80.80 | 75.44 |
| | GPT-4o | 87.24 | 64.56 | 70.21 | 63.14 |
| | Claude-4-Sonnet | 95.04 | 83.27 | 83.84 | 76.57 |
| *LLM-as-a-Judge: ICAI* | | | | | |
| | Qwen3-8B | 93.13 | 79.73 | 88.24 | 76.71 |
| | Qwen3-32B | 93.37 | 82.57 | 87.37 | 70.85 |
| | GPT-4o | 89.61 | 75.01 | 76.79 | 63.43 |
| | Claude-4-Sonnet | 94.57 | 86.22 | 89.34 | 80.86 |
| ***Auto-Rubric (Ours) - HelpSteer3*** | | | | | |
| | Qwen3-8B | 93.50 | 80.91 | 88.28 | 75.71 |
| | Qwen3-32B | 93.80 | 82.27 | 88.11 | 80.86 |
| | GPT-4o | 92.09 | 78.56 | 76.83 | 69.71 |
| | Claude-4-Sonnet | **95.81** | 86.90 | **89.49** | 83.18 |
| ***Auto-Rubric (Ours) - UltraFeedback*** | | | | | |
| | Qwen3-8B | 93.10 | 80.54 | 88.60 | 75.43 |
| | Qwen3-32B | 93.03 | 80.69 | 87.50 | 79.14 |
| | GPT-4o | 90.42 | 79.00 | 76.40 | 65.71 |
| | Claude-4-Sonnet | 95.04 | **87.90** | 87.50 | **81.71** |

**Bold** indicates the best score in each column across all methods.
Scores marked with '–' are unavailable from original publications.
Complete results with all models and variants are provided in Table 9.

natural stopping criterion prevents rubric inflation while ensuring comprehensive coverage of the evaluation space. Full hyperparameter settings are summarized in Appendix B.

## 4.4 ABLATION STUDIES

We conduct ablation studies to isolate the contribution of each core component of our framework, detailed in Table 2: (1) the iterative refinement of query-specific rubrics, (2) the information-theoretic selection of rubric subsets, (3) the final hierarchical structuring of the rubrics.

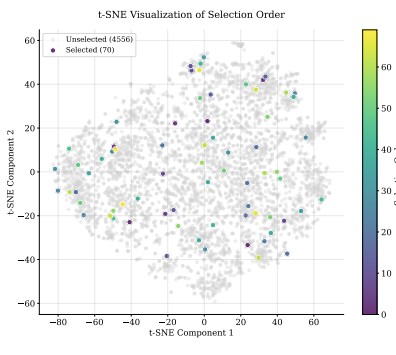 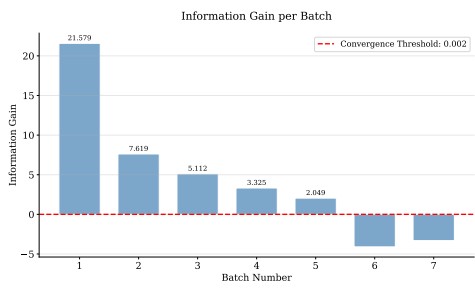

(a) t-SNE Visualization of Selection Order      (b) Information Gain per Batch

Figure 3: Analysis of the information-theoretic rubric selection process. (a) Early selections (darker points) maximize diversity by covering distinct semantic clusters. (b) Information gain (coding rate increment) diminishes as the set grows.

The full iterative **Propose-Evaluate-Revise** process outperforms single-pass generation by +2.43% on RewardBench2, confirming the necessity of validation-driven refinement. Our information-theoretic selection strategy surpasses random selection by +3.16% on RewardBench2, demonstrating the importance of diversity-aware selection. The Theme-Tips hierarchical structure improves accuracy by +1.13% on RewardBench2 over a flat list, demonstrating the value of balancing general criteria (Theme) with specific guidance (Tips).

Table 2: Comprehensive ablation studies across framework components.

| Component/Method | RewardBench2 | RM-Bench |
|---|---|---|
| *Iterative Refinement Components* | | |
| Single-pass Generation | 79.84 | 86.07 |
| Blind Revision (No Failed Rubric) | 81.98 (+2.14) | 85.79 (-0.28) |
| Our Method (Full Iterative) | 82.27 (+2.43) | 88.11 (+2.04) |
| *Rubric Selection Methods* | | |
| Random Selection | 79.11 | 86.80 |
| Our Method (Coding Rate) | 82.27 (+3.16) | 88.11 (+1.31) |
| *Rubric Structure Variants* | | |
| No Special Structure | 81.14 | 87.41 |
| General (with optional rubrics) | 80.01 (-1.13) | 86.28 (-1.13) |
| Theme (no tips) | 80.77 (-0.37) | 87.59 (+0.18) |
| Theme-Tips | 82.27 (+1.13) | 88.11 (+0.70) |

## 5 ANALYSIS OF CORE RUBRICS

To validate that our framework produces high-quality, interpretable evaluation criteria, we apply the analysis framework defined in Section 3.4 to the final extracted rubric set. This allows us to quantify the utility of each rubric and demonstrate that the final set is composed of complementary, non-redundant rubrics.

As shown in Table 3, foundational rubrics like "Prioritize clarity" exhibit extremely high coverage (97.92%) and contribution (7.09% accuracy drop if removed), acting as the basis of the evaluation. In contrast, specialized rubrics like "Ensure narrative fidelity" have lower coverage (71.91%) but the highest precision (68.24%), effectively handling niche scenarios that broader rubrics might miss. The significant contribution score of every rubric validates that our information-theoretic selection successfully produces a non-redundant set where each element plays a critical role. This analysis confirms that we are not just generating rubrics, but high-quality, structured evaluation knowledge. Complete rubric collections extracted from different datasets are presented in Appendix J.

Table 3: Analysis of the final rubrics, demonstrating their individual value.

| Rubric Theme | Coverage (%) | Precision (%) | Contribution ($\Delta$ Acc %) |
|---|---|---|---|
| Ensure factual accuracy. | 91.91 | 62.78 | 4.42 |
| Maintain strict adherence. | 85.90 | 59.16 | 3.72 |
| Prioritize clarity. | 97.92 | 65.07 | 7.09 |
| Deliver comprehensive. | 97.16 | 65.92 | 4.78 |
| Ensure narrative fidelity. | 71.91 | 68.24 | 3.68 |

## 5.1 APPLICATION TO POLICY OPTIMIZATION

To validate the practical utility of our extracted rubrics beyond evaluation benchmarks, we apply them to downstream policy optimization via DPO (Rafailov et al., 2024). Results are shown in Table 4.

Table 4: DPO performance comparison on Arena-Hard and AlpacaEval. We report win-rates (%) against the baseline. **Style-Ctrl** refers to Style-Controlled win rates for Arena-Hard and Length-Controlled win rates for AlpacaEval. Baseline results (†) are sourced from Viswanathan et al. (2025).

| Model / Strategy | Arena-Hard | | AlpacaEval | |
|---|---|---|---|---|
| | Vanilla | Style-Ctrl | Vanilla | Style-Ctrl |
| GPT-4-0314 (Reference)[†] | 50.0 | 50.0 | 22.1 | 35.3 |
| Qwen2.5-7B-Instruct (Base)[†] | 51.3 | 42.8 | 33.5 | 36.2 |
| + SFT (Distilled)[†] | 32.6 | 29.2 | 36.1 | 33.3 |
| + DPO (UltraFeedback)[†] | 52.8 | 47.9 | 33.7 | 38.7 |
| + DPO (AI Judge)[†] | 51.0 | 44.4 | 28.8 | 33.4 |
| + DPO (ArmoRM)[†] | 50.8 | 46.4 | 37.6 | 38.1 |
| + DPO (Skywork)[†] | **55.1** | 50.3 | **44.8** | 41.5 |
| + DPO (RLCF)[†] | 54.6 | 48.4 | 36.2 | 37.1 |
| + DPO (**Auto-Rubric**, Ours) | 54.2 | **57.0** | 42.8 | **42.2** |

On vanilla metrics, Skywork achieves the highest scores (55.1% on Arena-Hard, 44.8% on AlpacaEval), with our method ranking second (54.2%, 42.8%). However, the gap between vanilla and style-controlled metrics reveals a critical distinction: Skywork exhibits a large negative gap (-4.8% on Arena-Hard, -3.3% on AlpacaEval), indicating that a substantial portion of its vanilla performance stems from generating longer responses. In contrast, our method shows a positive gap (+2.8% on Arena-Hard, +0.7% on AlpacaEval), achieving higher style-controlled scores (57.0%, 42.2%) than all baselines. This suggests that our policy produces genuinely higher-quality responses rather than exploiting length as a proxy for quality. The trade-off reflects a deliberate design choice: our rubrics (e.g., "Prioritize Clarity," "Strict Adherence") explicitly discourage verbosity, leading to more concise but substantive outputs. For applications where response quality matters more than superficial length, our rubric-guided approach offers a principled alternative to reward models that may inadvertently encourage verbose responses.

## 6 CONCLUSION

We introduced Auto-Rubric, a novel, training-free framework that successfully addresses the critical trade-off between performance, data efficiency, and interpretability in reward modeling. Our work demonstrates that the core criteria underlying human preferences can be automatically distilled into a compact, generalizable, and non-redundant set of "Theme-Tips" rubrics. The efficacy of this approach is notable: using just 70 preference pairs (1.5% of the source data), our extracted rubrics enabled a Qwen3-8B model to outperform specialized, fully-trained reward models, setting a new state-of-the-art for training-free methods on RewardBench2. These results provide strong evidence that by shifting the focus from opaque **reward model learning** to transparent **rubric learning**, we can forge a more scalable, efficient, and trustworthy path for LLM alignment.

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

## A  THE USE OF LARGE LANGUAGE MODELS

In the preparation of this manuscript, we leveraged several large language models—including Google's Gemini, Alibaba's Qwen, and Anthropic's Claude—to assist with language editing and textual refinement. The role of these models was strictly limited to enhancing the clarity, grammatical correctness, fluency, and stylistic consistency of the manuscript. Specific tasks included refining sentence structure, suggesting alternative phrasings for improved readability, and harmonizing terminology and tone across sections. All outputs generated or suggested by these models were carefully evaluated, critically revised, and ultimately approved by the authors. The authors retain full responsibility for the scientific content, accuracy, and integrity of the final manuscript.

## B  ALGORITHM HYPERPARAMETERS

To ensure reproducibility, we provide detailed definitions of all hyperparameters used in our framework. These parameters control the two-level optimization process described in Section 4.3.

Table 5: Hyperparameters used in the Auto-Rubric framework.

| Symbol | Value | Description |
|---|---|---|
| *Outer Loop: Global Rubric Selection* | | |
| $B$ | 10 | **Mini-batch size.** Number of preference pairs processed in each selection iteration. |
| $\varepsilon$ | 0.5 | **Distortion threshold.** Controls redundancy tolerance in the coding rate objective (Eq. 6). Higher values penalize similar rubrics more aggressively. |
| $\tau_{\min}$ | 0.002 | **Convergence threshold.** Minimum marginal coding rate gain required to continue selection. |
| $p_{\text{patience}}$ | 2 | **Patience.** Number of consecutive iterations with marginal gain $< \tau_{\min}$ tolerated before early stopping. |
| *Inner Loop: Query-Specific Rubric Refinement* | | |
| $E_{\max}$ | 10 | **Maximum refinement iterations.** Maximum number of Propose-Evaluate-Revise cycles for a single preference pair. |
| $k$ | 5 | **Final rubric count.** Target number of Theme-Tips rubrics in the final structured set. |

**Note on $E_{\max}$:** Unlike training epochs in deep learning, $E_{\max}$ refers to self-correction iterations for a single sample, not passes over the entire dataset. As shown in Figure 5, convergence typically occurs within 3-5 iterations.

**Note on $\varepsilon$:** This parameter controls the "negative information gain" phenomenon in Figure 3b. As $|R_{\text{core}}|$ increases, the penalty term $\frac{1}{\varepsilon^2 |R_{\text{core}}|}$ in Eq. 6 eventually outweighs semantic expansion from redundant rubrics, triggering early termination.

## C  EXPERIMENT SETTING DETAILS

**Evaluation Protocol.**  For all benchmarks, we use accuracy as the primary metric. To ensure robust results, we employ voting strategies tailored to each benchmark's characteristics: voting@10 for RewardBench2, voting@5 for RewardBench and JudgeBench, and voting@1 for RM-Bench. These strategies balance result reliability with computational efficiency. A comprehensive test-time scaling analysis examining the trade-offs between voting numbers and performance is presented in Appendix F.

**Inference Configuration.** For open-source Qwen models, we use the following batch inference parameters: temperature = 0.7, top-p = 0.8, top-k = 20, with think mode enabled for all stages of rubric construction and evaluation. For proprietary models (GPT-4o, Claude-4-Sonnet), we use their default API configurations.

**Policy Optimization.** We adopt the DPO setup corresponding to the main-text results in Section 5.1. Concretely, we finetune **Qwen2.5-7B-Instruct** using Direct Preference Optimization (DPO) (Rafailov et al., 2024) on **WildChat** (Zhao et al., 2024) prompts labeled by our Auto-Rubric evaluator (HelpSteer3 rubrics), retaining only high-confidence (5/5) pairs. Training is performed for 2 epochs with a global batch size of 1024 and maximum sequence length 2048 on two 8×H20 nodes. We compare against the DPO baselines of Viswanathan et al. (2025), and report win rates on **Arena-Hard** (Li et al., 2024a) and **AlpacaEval** (Dubois et al., 2025) in Table 4.

# D   ANALYSIS ON THE GENERALIZABILITY OF MODEL-GENERATED RUBRICS

To select the optimal LLM for our framework, we analyzed the generalizability of evaluation rubrics generated by three leading models: **Qwen3-32B, GPT-4o, and Claude-4-Sonnet**. We benchmarked each model's performance as an evaluator, both in a baseline condition (no rubric) and when guided by the rubrics from each of the three generators. The results in Figure 4 reveal clear patterns in rubric quality and cross-model utility.

The findings confirm two main points. First, in all scenarios, applying a model-generated rubric provides a significant performance uplift over the baseline. Second, and more critically, **Qwen3-32B generated rubrics exhibit the strongest generalizability**. This is most evident in the cross-model tests; for example, Qwen3-32B's rubric boosts GPT-4o's performance on RewardBench2 to **0.7902** and significantly higher than the score achieved with its own rubric (0.7453). While Claude-4-Sonnet consistently posts the highest absolute scores, proving it is a powerful standalone evaluator, the superior and consistent performance uplift that **Qwen3-32B's rubrics provide to *other* models** makes it the unambiguous choice for generating a robust, universally applicable set of rubrics for our main experiments.

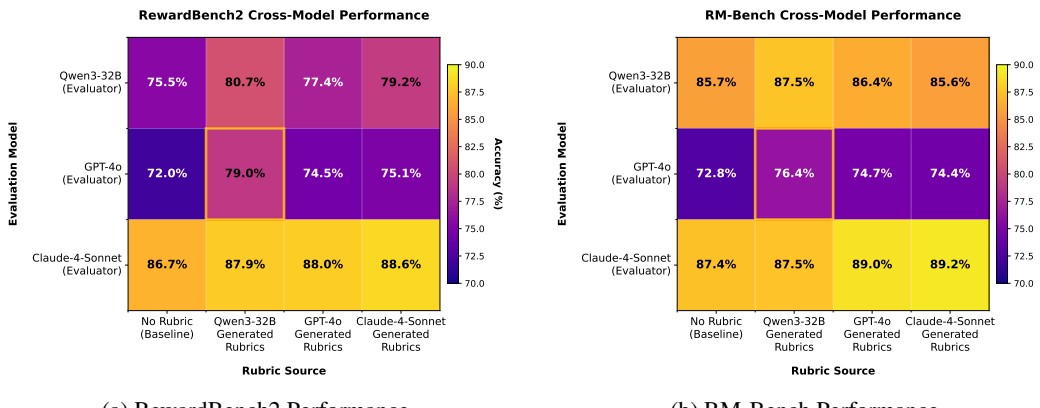

(a) RewardBench2 Performance                    (b) RM-Bench Performance

Figure 4: Cross-model rubric generalization analysis. The heatmaps show performance when evaluation models use rubrics generated by different LLMs. The orange borders highlight the best cross-model transferability: GPT-4o evaluator with Qwen3-32B rubrics achieves 79.02% on RewardBench2 and 76.37% on RM-Bench, demonstrating superior generalizability of Qwen3-32B generated rubrics.

# E   QUERY-SPECIFIC ACCURACY IMPROVEMENT ANALYSIS

To further understand the learning dynamics of our rubric extraction framework, we analyze the query-specific accuracy improvement across training epochs for both datasets used in our exper-

iments. Figure 5 illustrates the progressive enhancement in accuracy as our iterative refinement process generates and refines rubrics.

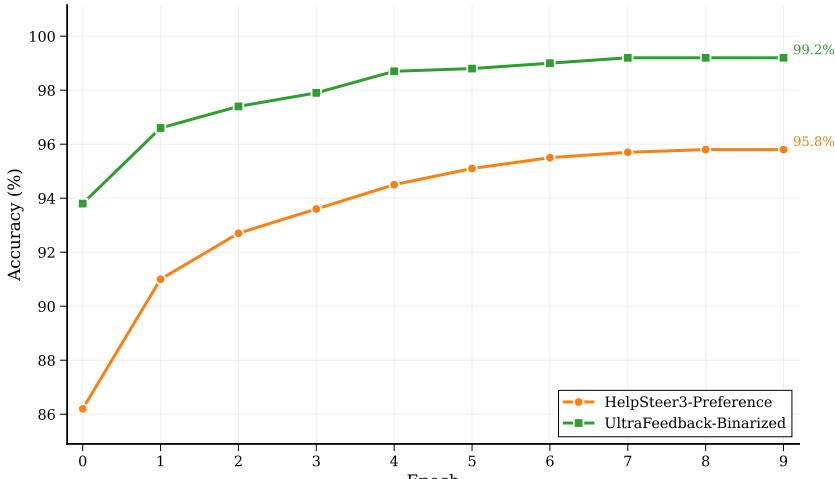

Figure 5: Query-specific generation accuracy improvement across epochs. The figure shows the progressive enhancement in accuracy for both HelpSteer3-Preference (orange line) and UltraFeedback-Binarized (green line) datasets. Both curves demonstrate rapid initial improvement followed by convergence to high accuracy levels, with UltraFeedback-Binarized achieving 99.20% and HelpSteer3-Preference reaching 95.80% by epoch 9. The steep initial gradient indicates the effectiveness of our iterative Propose-Evaluate-Revise mechanism in quickly identifying and refining core evaluation rubrics.

The results reveal several key insights about our framework's learning dynamics:

**Rapid Initial Convergence.** Both datasets exhibit steep accuracy improvements in the first 2-3 epochs, with HelpSteer3-Preference jumping from 86.1% to 92.7% (epoch 0 to 2) and UltraFeedback-Binarized improving from 93.9% to 97.4%. This rapid initial improvement demonstrates the effectiveness of our iterative refinement process in quickly identifying fundamental evaluation rubrics that govern human preferences.

**Dataset-Specific Characteristics.** UltraFeedback-Binarized consistently achieves higher accuracy levels and faster convergence, reaching 99.20% by epoch 9 compared to HelpSteer3-Preference's 95.80%. This difference likely reflects the distinct annotation methodologies: Ultra-Feedback's GPT-4-based scoring may exhibit more consistent patterns compared to HelpSteer3's human annotations, which naturally contain more subjective variance.

**Convergence Stability.** Both curves demonstrate saturation behavior after epoch 6, with minimal improvements in subsequent iterations. This validates our adaptive stopping mechanism and suggests that the core evaluation rubrics underlying human preferences can be effectively captured within a limited number of refinement cycles.

**Cross-Dataset Validation.** The consistent improvement patterns across both datasets support our core hypothesis about rubric convergence—despite different domains, annotation methods, and preference distributions, the underlying evaluation rubrics exhibit similar optimization dynamics, confirming the generalizability of our approach.

## F    TEST-TIME SCALING ANALYSIS

To evaluate the robustness and stability of our rubric-based evaluation framework, we investigate how performance scales with increasing voting numbers during test-time inference on Reward-

Bench2. This analysis provides crucial insights into the trade-offs between computational cost and evaluation reliability.

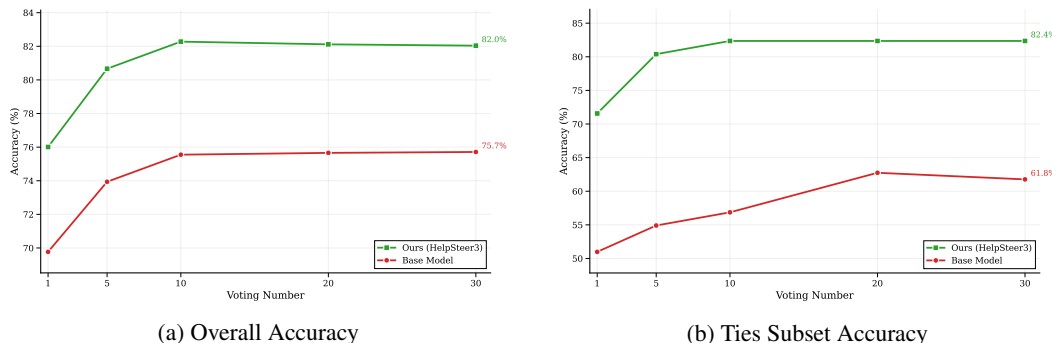

(a) Overall Accuracy            (b) Ties Subset Accuracy

Figure 6: Test-time scaling analysis on RewardBench2 benchmark. (a) Overall accuracy improvement shows our rubric-enhanced HelpSteer3-Preference approach (green line) consistently outperforming the base model (red line) by 6-7 percentage points across all voting strategies (1, 5, 10, 20, 30). (b) Ties subset accuracy focuses on challenging cases where the base model produces tie decisions, demonstrating dramatic improvements of approximately 20 percentage points, highlighting the critical role of explicit rubrics in resolving ambiguous preference decisions.

The test-time scaling analysis reveals several important characteristics of our framework:

**Consistent Performance Advantage.** Figure 6a demonstrates that our rubric-enhanced approach maintains a consistent 6-7 percentage point advantage over the base model across all voting strategies. This systematic improvement suggests that our extracted rubrics provide fundamental evaluation capabilities that are orthogonal to the benefits of ensemble voting, creating additive performance gains.

**Rapid Convergence with Low Voting Numbers.** Both approaches show the most significant improvements when scaling from voting@1 to voting@5, with diminishing returns thereafter. This pattern indicates that the primary benefits of ensemble voting can be captured with relatively modest computational overhead. For practical deployment, voting@5 to voting@10 appears to offer the optimal balance between performance and efficiency.

**Superior Performance on Challenging Cases.** Figure 6b provides particularly compelling evidence for our framework's effectiveness. On the ties subset—representing the most challenging evaluation scenarios where base models struggle to make decisive judgments—our rubric-enhanced approach shows dramatic improvements of approximately 20 percentage points. This substantial gap highlights the critical role of explicit rubrics in providing discriminative power precisely where it is most needed.

**Plateau Behavior and Computational Efficiency.** Both figures demonstrate plateau behavior beyond voting@10, suggesting that additional computational investment yields marginal returns. This finding has important practical implications: our framework achieves near-optimal performance with moderate ensemble sizes, making it computationally efficient for real-world deployment while maintaining high evaluation quality.

**Robustness Across Difficulty Levels.** The consistent performance patterns across both overall accuracy and ties subset accuracy indicate that our rubrics provide robust evaluation capabilities that scale effectively across different difficulty levels. This robustness is crucial for practical applications where evaluation systems must handle diverse query types and ambiguous cases reliably.

# G    DETAILED EXPERIMENTAL ANALYSIS

To provide comprehensive insights into our framework's effectiveness, we conduct detailed analyses across multiple benchmarks and evaluation dimensions. This section examines where our rubric-guided approach provides the most significant value, focusing on challenging evaluation scenarios and domain-specific performance patterns.

## G.1    CROSS-BENCHMARK PERFORMANCE ANALYSIS

Our detailed analysis encompasses two complementary benchmarks that together provide a comprehensive view of rubric effectiveness: RM-Bench, which allows us to examine performance on samples of varying difficulty levels, and RewardBench2, which offers diverse evaluation dimensions including challenging edge cases.

**RM-Bench: Difficulty-Stratified Analysis.**    We conduct a stratified analysis on RM-Bench to understand how our rubrics perform across different difficulty levels (Table 6). The results reveal a clear and consistent pattern: our rubrics excel at resolving the most challenging cases, where base models struggle to make accurate preference judgments.

The difficulty-stratified analysis shows that hard samples benefit substantially more from rubric guidance (+4.68%) compared to the overall improvement (+2.45%). This 2x amplification effect on difficult cases demonstrates that our rubrics provide crucial discriminative power precisely where it is most needed—in scenarios where implicit evaluation rubrics are insufficient.

Domain-specific patterns further illuminate our framework's targeted strengths. The **Chat** domain exhibits the most dramatic improvement (+13.95% on hard samples), highlighting our rubrics' effectiveness in the notoriously subjective area of conversational evaluation where nuanced judgment rubrics are critical. Substantial gains are also observed in **Math** (+4.54%) and **Safety-Refuse** (+3.64%), demonstrating broad applicability across diverse reasoning and safety scenarios.

Table 6: Performance analysis on RM-Bench using Qwen3-32B across domains and difficulty levels. All values are accuracies (%). $\Delta$ denotes the improvement.

| Domain | Overall (%) | | | Hard Samples (%) | | |
|---|---|---|---|---|---|---|
| | Base | Rubric | $\Delta$ | Base | Rubric | $\Delta$ |
| **Overall** | 85.67 | **88.12** | +2.45 | 77.74 | **82.42** | +4.68 |
| Chat | 70.37 | **77.00** | +6.63 | 36.95 | **50.90** | +13.95 |
| Math | 89.29 | **91.89** | +2.60 | 85.44 | **89.98** | +4.54 |
| Code | 74.90 | **76.95** | +2.05 | 72.08 | **73.54** | +1.46 |
| Safety-Refuse | 94.68 | **96.64** | +1.96 | 91.55 | **95.19** | +3.64 |
| Safety-Response | 85.42 | 85.35 | -0.07 | 68.58 | **72.61** | +4.03 |

**RewardBench2: Evaluation Dimension Analysis.**    To complement our difficulty-focused RM-Bench analysis, we examine performance across diverse evaluation dimensions on RewardBench2 (Table 7). RewardBench2 provides a more challenging and comprehensive evaluation setting, allowing us to understand where rubric-guided evaluation provides the most significant advantages across different types of evaluation rubrics.

The results reveal consistent and substantial improvements across all evaluation dimensions, with our rubrics achieving a remarkable overall improvement of +6.72% (from 75.55% to 82.27%). This substantial gain on a challenging benchmark demonstrates the robust effectiveness of our framework across diverse evaluation scenarios.

The most notable finding is the significant improvement on the **Ties** subset (+25.49%), jumping from 56.86% to 82.35%. This substantial gain represents the most challenging evaluation scenarios—where base models struggle to make decisive judgments—and highlights the critical discriminative power that explicit rubrics provide in ambiguous cases. The **Safety** domain also shows significant enhancement (+10.34%), demonstrating our rubrics' effectiveness in navigating nuanced safety considerations that require careful balance between multiple competing factors.

Importantly, even domains where base models already achieve strong performance show meaningful improvements: **Factuality** gains +8.84% and **Precise IF** improves by +5.62%. This pattern indicates that our rubrics provide value across the full spectrum of evaluation difficulty, from challenging edge cases to well-established domains, confirming the broad applicability and robustness of our approach.

Table 7: Performance analysis on RewardBench2 using Qwen3-32B across evaluation dimensions. All values are accuracies (%). $\Delta$ denotes the improvement.

| Dimension | Base (%) | Rubric (%) | $\Delta$ (%) |
|-----------|----------|------------|--------------|
| **Overall** | 75.55 | **82.27** | +6.72 |
| Precise IF | 46.88 | **52.50** | +5.62 |
| Ties | 56.86 | **82.35** | +25.49 |
| Factuality | 65.68 | **74.53** | +8.84 |
| Focus | 92.93 | **93.13** | +0.20 |
| Safety | 77.13 | **87.47** | +10.34 |
| Math | 85.79 | **86.34** | +0.55 |

## H    MULTI-SEED STABILITY ANALYSIS

To rigorously validate the stability of our approach and address concerns regarding the inherent stochasticity of LLM generation and evaluation, we conducted comprehensive multi-seed experiments: We ran the entire rubric extraction pipeline 5 independent times using different random seeds, sampling distinct subsets of preference data from HelpSteer3. Each run produced a different rubric set, which we then evaluated across all four benchmarks. We repeated the baseline evaluation 5 times to account for inference-time stochasticity in the LLM's zero-shot judgments.

Table 8 reports the mean and standard deviation across the 5 runs. The results demonstrate high stability and statistical significance across three key dimensions:

**Baseline Stability.**    The Base models exhibit very low variance (SD $\approx 0.1\%$), confirming that the evaluation harness itself is stable and that inference-time stochasticity has minimal impact on zero-shot performance.

**Rubric Stability.**    While "Ours" has slightly higher variance (SD $\approx 0.3\% - 0.4\%$) due to the data sampling process in rubric extraction, this variance is negligible compared to the performance gains. The standard deviations remain consistently small across all benchmarks (typically $< 0.45$), indicating that our Propose-Evaluate-Revise mechanism reliably converges to high-quality rubrics regardless of the specific seed data sampled.

**High Signal-to-Noise Ratio.**    Crucially, the performance gain ($\Delta$) systematically exceeds the variance of both the Base and Our method. On RewardBench2 (Qwen3-32B), the gain (+6.53%) is approximately $19\times$ the standard deviation (0.35%). On RM-Bench (Qwen3-32B), the gain (+2.34%) is over $10\times$ the standard deviation (0.22%). Even the minimum score recorded across all 5 runs of "Ours" consistently outperforms the maximum score of the "Base" runs, demonstrating that the improvements are systematic and statistically significant rather than artifacts of random sampling or LLM stochasticity.

## I    COMPLETE BENCHMARK RESULTS

## J    EXTRACTED RUBRIC COLLECTIONS

This section presents the complete sets of query-agnostic rubrics extracted by our framework from different datasets and experimental configurations. These rubrics demonstrate the structured "Theme-Tips" hierarchy that emerges from our information-theoretic selection and thematic induction processes.

Table 8: Multi-seed stability analysis. We report Mean ± Standard Deviation across 5 independent runs with different random seeds for both Base and Ours (HelpSteer3). The low variance and high signal-to-noise ratio confirm the robustness of our approach.

| Model | Method | RewardBench | RewardBench2 | RM-Bench | JudgeBench |
|---|---|---|---|---|---|
| Qwen3-32B | Base (5 Runs) | 92.98 ± 0.08 | 75.62 ± 0.15 | 85.71 ± 0.11 | 75.68 ± 0.13 |
| | Ours (HelpSteer3, 5 Runs) | 93.75 ± 0.18 | 82.15 ± 0.35 | 88.05 ± 0.22 | 80.72 ± 0.31 |
| Qwen3-8B | Base (5 Runs) | 92.89 ± 0.09 | 74.42 ± 0.18 | 86.85 ± 0.12 | 73.18 ± 0.14 |
| | Ours (HelpSteer3, 5 Runs) | 93.42 ± 0.21 | 80.81 ± 0.38 | 88.15 ± 0.25 | 75.65 ± 0.29 |
| GPT-4o | Base (5 Runs) | 88.28 ± 0.15 | 72.08 ± 0.21 | 72.85 ± 0.18 | 68.35 ± 0.22 |
| | Ours (HelpSteer3, 5 Runs) | 91.95 ± 0.25 | 78.40 ± 0.45 | 76.75 ± 0.33 | 69.50 ± 0.41 |

## J.1 HELPSTEER3-PREFERENCE DATASET RUBRICS

The following rubrics were extracted from the HelpSteer3-Preference dataset:

### Theme 1: Factual Accuracy and Canonical Consistency

**Theme:** Ensure factual accuracy, canonical consistency, and avoid fabrication or hallucination in responses.
- **Tip 1:** For queries about *Undertale*, ensure all character motivations and gameplay mechanics align with established lore, avoiding speculative or contradictory claims.
- **Tip 2:** When discussing historical milestones like early synchronized sound cartoons, correctly attribute "Steamboat Willie" instead of "My Old Kentucky Home" to maintain reliability.
- **Tip 3:** In responses involving *Hogwarts* students, include only canonically portrayed students with academically accurate achievements, excluding professors or non-student figures.
- **Tip 4:** Avoid inventing Sumerian texts or fabricated survey links; instead, acknowledge missing context and request clarification when necessary, especially for niche cultural references.

### Theme 2: Strict Adherence to Prompt Requirements

**Theme:** Maintain strict adherence to prompt structure, formatting, and explicit user requirements.
- **Tip 1:** When asked for a single word, provide exactly one word without redundancy or additional suggestions, as in responses requiring minimal output.
- **Tip 2:** For prompts specifying 100 items, deliver a complete list even if the topic is broad, proactively selecting a relevant subject to fulfill the quantitative requirement.
- **Tip 3:** In tagline creation, directly incorporate core technology benefits like "distance at impact" and avoid vague or redundant phrasing that dilutes product relevance.
- **Tip 4:** When the prompt requires the word "scenery" followed by a colon and a one-word term, follow this exact syntactic structure without deviation.

### Theme 3: Clarity and Structured Organization

**Theme:** Prioritize clarity, conciseness, and structured organization to enhance readability and directness.

- **Tip 1:** For a "Thank you" prompt, respond with a concise acknowledgment and an open invitation for further questions, avoiding assumptions about the user being a student or lawyer.
- **Tip 2:** When summarizing steps for building a dropshipping agent business, use bullet points or numbered lists to present key points logically and avoid hallucinated information.
- **Tip 3:** In audit findings related to deposit insurance boards, structure responses with precise, actionable items and conclude with a concise summary emphasizing implications.
- **Tip 4:** Avoid excessive formatting like bold text or unnecessary punctuation when explaining grammatical correctness, maintaining a straightforward and professional tone.

### Theme 4: Comprehensive and Detailed Analysis

**Theme:** Deliver comprehensive, detailed, and thematically coherent narratives or analyses that fully address all prompt elements.
- **Tip 1:** For a CFA Institute Investment Foundations® Certificate explanation, include curriculum, eligibility, exam format, preparation resources, benefits, and continuing education with specific examples.
- **Tip 2:** In a fantasy story response, incorporate rich narrative detail, distinct character development, and immersive world-building such as vivid settings and dynamic interactions.
- **Tip 3:** When addressing a tax-proportional legislature, outline mechanics, implications, data collection, representation quotas, equity concerns, and constitutional considerations comprehensively.
- **Tip 4:** For a horror anime scene, use INT./EXT. designations, emphasize atmospheric tension, and describe creature details like a rhombus tail and chameleon-like head to align with anime style.

### Theme 5: Narrative and Contextual Fidelity

**Theme:** Ensure narrative and contextual fidelity by preserving character dynamics, tone, and worldbuilding consistency.
- **Tip 1:** In responses involving Jade's character, maintain her authoritative yet professional tone, avoiding hostile shifts that contradict established behavior.
- **Tip 2:** For stories featuring Emily from KikoRiki, preserve her role as a mischievous prankster and integrate the whimsical tone when describing her failed morph into Rosa and the orange rear end mishap.
- **Tip 3:** When continuing a narrative about diaper use over potty training, maintain a playful, child-friendly tone and avoid contradictions with the original theme.
- **Tip 4:** In therapeutic role-play scenarios, prioritize immersive engagement with the patient's imaginative world through dialogue and validation, rather than clinical checklists.

## J.2 ULTRAFEEDBACK-BINARIZED DATASET RUBRICS

The following rubrics were extracted from the UltraFeedback-Binarized dataset:

### Theme 1: Factual Accuracy and Domain-Specific Knowledge

**Theme:** The answer must be factually accurate and grounded in correct domain-specific knowledge, avoiding misconceptions, logical errors, or speculative assumptions.
- **Tip 1:** Correctly apply scientific, technical, or mathematical principles (e.g., gravity, regex syntax, Pig Latin rules) with precision.
- **Tip 2:** Avoid perpetuating false premises (e.g., birds producing seeds) and instead clarify biological or conceptual inaccuracies.

- **Tip 3:** Use verified data, proper citations, and accurate terminology (e.g., Azure work-flows, MLA formatting, product design details).
- **Tip 4:** When faced with ambiguity, seek clarification rather than making unfounded assumptions.
- **Tip 5:** Preserve original information in translations without adding, omitting, or distorting meaning.

### Theme 2: Explicit Requirement Fulfillment

**Theme:** The answer must directly fulfill the user's explicit requirements in structure, content, and format, adhering strictly to all stated constraints.
- **Tip 1:** Follow prescribed structural elements (e.g., opening phrases, question framing, section order).
- **Tip 2:** Respect formatting rules (e.g., LaTeX, APA, SQL schema limits, phone number patterns).
- **Tip 3:** Address every component of multi-part queries (e.g., examples, explanations, code, citations).
- **Tip 4:** Use only valid functions, libraries, or commands within the correct technical context (e.g., Streamlit, PL/pgSQL).
- **Tip 5:** Extract or generate responses using only permitted sources (e.g., exact text spans, background passages).

### Theme 3: Clarity and Logical Organization

**Theme:** The answer must provide clarity, coherence, and completeness through well-structured, concise, and logically organized reasoning.
- **Tip 1:** Offer step-by-step explanations that make reasoning transparent and verifiable.
- **Tip 2:** Maintain grammatical correctness and preserve original language or formatting conventions.
- **Tip 3:** Avoid unnecessary elaboration, redundancy, or irrelevant details that distract from the core task.
- **Tip 4:** Ensure responses are self-contained and understandable without external context.
- **Tip 5:** Use precise connectors and descriptive language to maintain fidelity in translation or interpretation.

### Theme 4: Depth and Contextual Relevance

**Theme:** The answer must demonstrate depth and richness by integrating specific examples, actionable strategies, and contextual relevance.
- **Tip 1:** Include concrete, scenario-specific illustrations (e.g., AR gameplay mechanics, cultural program metrics).
- **Tip 2:** Provide practical implementation guidance with technical detail (e.g., iOS frameworks, OpenGL code).
- **Tip 3:** Link abstract concepts to real-world applications (e.g., symbolism in literature, ESG factors in market entry).
- **Tip 4:** Show progression or transformation (e.g., habit formation plans, historical scientific impact).
- **Tip 5:** Balance breadth and depth by covering multiple dimensions while offering nuanced analysis.

**Theme 5: Ethical Responsibility and User Alignment**

**Theme:** The answer must prioritize ethical responsibility, user alignment, and functional utility in its approach and tone.

- **Tip 1:** Reframe potentially offensive or harmful terms proactively to maintain respectful communication.

- **Tip 2:** Focus on actionable solutions rather than dismissive or overly theoretical responses.

- **Tip 3:** Tailor advice to the user's role, goals, or identity (e.g., UK lawyer, developer, educator).

- **Tip 4:** Encourage engagement through clear invitations or follow-up prompts when interaction is intended.

- **Tip 5:** Enhance transparency with confidence indicators or explicit justifications for conclusions.

Table 9: Complete Performance of Models on Four Key Benchmarks (in Percent). This table extends Table 1 with all model sizes and additional baselines.

| Method Type | Model / Variant | RewardBench | RewardBench2 | RM-Bench | JudgeBench |
|---|---|---|---|---|---|
| *Zero-Shot Base Models* | | | | | |
| | Qwen3-8B | 92.93 | 74.37 | 86.90 | 73.14 |
| | Qwen3-14B | 92.66 | 76.30 | 87.70 | 75.14 |
| | Qwen3-32B | 92.96 | 75.55 | 85.67 | 75.71 |
| | Qwen3-235B | 93.70 | 83.78 | 87.55 | 83.14 |
| | GPT-4o | 88.24 | 72.00 | 72.80 | 68.29 |
| | Claude-4-Sonnet | 94.61 | 86.70 | 85.70 | 78.29 |
| *In-Context Learning (k=5)* | | | | | |
| | Qwen3-8B | 90.18 | 72.57 | 86.83 | 67.71 |
| | Qwen3-14B | 89.58 | 74.89 | 87.29 | 70.86 |
| | Qwen3-32B | 90.82 | 75.24 | 85.91 | 74.00 |
| | Qwen3-235B | 90.42 | 81.38 | 86.91 | 82.86 |
| | GPT-4o | 88.89 | 73.11 | 74.02 | 68.91 |
| | Claude-4-Sonnet | 94.82 | 84.89 | 83.29 | 77.61 |
| *Training-based Reward Models* | | | | | |
| | ArmoRM-Llama3-8B-v0.1 | 90.40 | 66.50 | 69.30 | 59.70 |
| | J1-Llama-8B | 85.70 | – | 73.40 | 42.00 |
| | J1-Llama-70B | 93.30 | – | 82.70 | 60.00 |
| | R3-QWEN3-8B-14K | 87.50 | – | 82.10 | – |
| | R3-QWEN3-14B-LORA-4K | 89.30 | – | 84.90 | – |
| | RM-R1-Qwen-Instruct-32B | 92.90 | – | 79.10 | – |
| | RM-R1-DeepSeek-Distill-Qwen-32B | 90.90 | – | 83.90 | – |
| | Skywork-Reward-V2-Qwen3-8B | 93.70 | 78.20 | 82.60 | 73.40 |
| *LLM-as-a-Judge: Arena-Hard* | | | | | |
| | Qwen3-8B | 85.63 | 78.93 | 85.88 | 78.57 |
| | Qwen3-14B | 81.41 | 84.02 | 87.07 | 79.14 |
| | Qwen3-32B | 89.95 | 83.22 | 87.09 | 71.43 |
| | Qwen3-235B | 94.44 | 86.11 | 90.50 | 82.86 |
| | GPT-4o | 89.08 | 74.10 | 79.56 | 68.86 |
| | Claude-4-Sonnet | 95.41 | 84.99 | 88.11 | 82.57 |
| *LLM-as-a-Judge: MT-Bench* | | | | | |
| | Qwen3-8B | 92.56 | 73.40 | 84.84 | 77.14 |
| | Qwen3-14B | 93.03 | 76.89 | 85.37 | 79.43 |
| | Qwen3-32B | 91.99 | 74.29 | 80.80 | 75.44 |
| | Qwen3-235B | 94.03 | 84.07 | 87.04 | 82.86 |
| | GPT-4o | 87.24 | 64.56 | 70.21 | 63.14 |
| | Claude-4-Sonnet | 95.04 | 83.27 | 83.84 | 76.57 |
| *LLM-as-a-Judge: ICAI* | | | | | |
| | Qwen3-8B | 93.13 | 79.73 | 88.24 | 76.71 |
| | Qwen3-14B | 93.33 | 83.43 | 84.87 | 80.57 |
| | Qwen3-32B | 93.37 | 82.57 | 87.37 | 70.85 |
| | Qwen3-235B | 94.34 | 87.45 | 89.67 | 85.14 |
| | GPT-4o | 89.61 | 75.01 | 76.79 | 63.43 |
| | Claude-4-Sonnet | 94.57 | 86.22 | 89.34 | 80.86 |
| ***Auto-Rubric (Ours) - HelpSteer3*** | | | | | |
| | Qwen3-8B | 93.50 | 80.91 | 88.28 | 75.71 |
| | Qwen3-14B | 93.74 | 81.66 | 83.15 | 79.71 |
| | Qwen3-32B | 93.80 | 82.27 | 88.11 | 80.86 |
| | Qwen3-235B | **95.81** | 86.46 | 89.51 | 85.43 |
| | GPT-4o | 92.09 | 78.56 | 76.83 | 69.71 |
| | Claude-4-Sonnet | **95.81** | 86.90 | 89.49 | 83.18 |
| ***Auto-Rubric (Ours) - UltraFeedback*** | | | | | |
| | Qwen3-8B | 93.10 | 80.54 | 88.60 | 75.43 |
| | Qwen3-14B | 93.67 | 80.91 | 88.72 | 78.86 |
| | Qwen3-32B | 93.03 | 80.69 | 87.50 | 79.14 |
| | Qwen3-235B | 94.54 | 85.97 | **89.58** | **86.29** |
| | GPT-4o | 90.42 | 79.00 | 76.40 | 65.71 |
| | Claude-4-Sonnet | 95.04 | **87.90** | 87.50 | 81.71 |

**Bold** indicates the best score in each column across all methods.
Scores marked with '–' are unavailable from original publications.

## K  PROMPT TEMPLATES

---

**Rubric Generation Prompt**

```
## Overview
You are an expert rubric writer for open-ended question. Your job
    is to
generate a self-contained set of evaluation criteria ("rubrics")
    for choosing a better answer from candidate answers to a given
    query. Rubrics can cover aspects such as factual correctness,
    depth of reasoning, clarity, completeness, style, helpfulness,
    and common pitfalls. Each rubric item must be fully self-
    contained so that non-expert readers need not consult any
    external information.

I will give you:
1. the query(maybe contains history messages)
2. candidate answers
3. which answer is better than others
4. critics by the human experts, and you need to carefully read the
     critics provided by human experts and summarize the rubrics.

NOTE: The number of rubrics should be LESS THAN OR EQUAL TO {number
    }

## Query
{query}

## Candidate Answers
<answer_1>{answer_1}</answer_1>
<answer_2>{answer_2}</answer_2>

## Better Answer
Answer {preference} is better than others.

## Critics
<critic>{critic}</critic>

## Output Format Requirements
<rubrics>your rubrics without index</rubrics>
```

---

Figure 7: Prompt for generating query-specific rubrics.

---

**Rubric Evaluation Prompt**

```
## Task Description
I will provide you with a set of rubrics, along with the current
    query and two responses. These rubrics are the primary basis for
     selecting the best answer. You must follow the steps specified
    in the Evaluation Process when conducting your evaluation
    process.

## Rubrics
{rubrics}

## Process
1. Confirm the task scenario of the current query and select the
    corresponding evaluation rubrics.
2. Identify the best response that meets the most selected rubrics.

## Query
{query}

## Response A
{response_a}

## Response B
{response_b}

## Output Requirement
Please choose the better response. Response "A", "B", or "tie"
    within the tags.
<preference>A/B/tie</preference>
```

Figure 8: Prompt for rubric-based pairwise evaluation.

**Rubric Revision Prompt**

```
## Overview
You are an expert rubric writer for open-ended question. A self-
    contained set of evaluation criteria ("rubrics") is needed for
    choosing a better answer from candidate answers to a given query
    . Since the rubrics generated in the previous round failed to
    correctly select a better answer, you need to revise the rubrics
    . rubrics can cover aspects such as factual correctness, depth
    of reasoning, clarity, completeness, style, helpfulness, and
    common pitfalls. Each rubric item must be fully self-contained
    so that non-expert readers need not consult any external
    information.

I will give you:
1. the query(maybe contains history messages)
2. candidate answers
3. which answer is better than others
4. critics by the human experts, and you need to carefully read the
     critics provided by human experts and summarize the rubrics.
5. previous round rubrics that should to be improved

NOTE: The number of rubrics should be LESS THAN OR EQUAL TO {number
    }

## Query
{query}

## Candidate Answers
<answer_1>
{answer_1}
</answer_1>

<answer_2>
{answer_2}
</answer_2>

## Better Answer
Answer {preference} is better than others.

## Previous Round rubrics
<rubric_1>
{previous_rubric_1}
</rubric_1>

## Output Format Requirements
Note: Ensure all outputs are placed within the tags like <tag>...</
    tag> as required!!!
<rubrics>
your improved rubrics without index
</rubrics>
```

Figure 9: Prompt for revising query-specific rubrics based on evaluation feedback.

---

**Rubric Structuring Prompt**

```
##Task Description
Your task is to generate a set of evaluation rubrics to identify
    the best answer, based on the suggestions for determining from
    the examples. I will give you some examples, and every example
    contains the query and suggestion which has been verified to
    help select the best answer.

## Requirements
- Rubrics must be fully self-contained so that non-expert readers
    need not consult any external information.
- Each rubric should assess an independent dimension and be non-
    contradictory with others.
- Rubrics ensure that the overall judgment remains aligned and
    consistent for all examples.
- The number of rubrics should be LESS THAN OR EQUAL TO 5. The
    number of tips for eachrubric should be LESS THAN OR EQUAL TO 5.
- Must strictly adhere to the Rubrics Format.

## Rubric Format
Each rubric consists of two parts:
- Theme: A concise and clear statement that captures the core focus
     of the rubric, and must be **necessary** for all queries with
    no assumption.
- Tips: Multiple bullet points that expand on or supplement the
    rubric and only focuses on some specific queries.

Here is an example of a rubric:
Theme: [Concise theme statement]
-Tip 1:
-Tip 2:
-Tip 3:
-(Optional: More tips as needed)

## Process
1. Based on the query and suggestions of each example, summarize
    the rubric of each example.
2. summarize the rubrics of each example, taking care to strictly
    adhere to the Requirements.

NOTE: The number of rubrics should be LESS THAN OR EQUAL TO 5. The
    number of tips for each rubric should be LESS THAN OR EQUAL TO
    5.

## Output Format Requirements
<rubrics>
Theme: [Concise theme statement]
-Tip 1: [Specific tip for certain queries]
-Tip 2: [Another specific tip]
-Tip 3: [Additional tip if needed]

Theme: [Another theme statement]
-Tip 1: [Related tip]
-Tip 2: [Another tip]
</rubrics>
```

Figure 10: Prompt for structuring the core rubric set into a "Theme-Tips" hierarchy.

