# OpenReview forum: "Auto-Rubric: Learning to Extract Generalizable Criteria for Reward Modeling"
_ICLR.cc/2026/Conference — ICLR 2026 Conference Desk Rejected Submission_

### Official Review · Reviewer_tuBg · 2025-10-28

**Soundness:** 1
**Presentation:** 3
**Contribution:** 2
**Rating:** 2
**Confidence:** 4

**Summary:**

This paper introduces Auto-Rubric, a training-free framework that learns an appropriate rubric "prompt" for downstream tasks. First, the framework performs a query-specific rubric generation using an iterative "Propose-Evaluate-Revise" pipeline based on a small number of preference pairs. Since this first step introduces a lot of redundancy, the framework further prunes this large pool greedily using an information-theoretic selection algorithm with a final rubric set from only 70 preference pairs (1.5% of the source data). The framework was evaluated using Qwen3-8B, Qwen3-14B, Qwen3-32B, and Qwen3-235B-Instruct-2507 on RewardBench, RewardBench2, JudgeBench, and RM-Bench.

**Strengths:**

**S1.** The paper introduces a training-free framework, which provides a scalable and accessible alternative to traditional reward modeling by achieving remarkable data efficiency.

**S2.** The framework is evaluated across a wide range of models and four benchmarks, supported by detailed ablation studies.

**S3.** The paper is easy to follow with a clear motivation.

**Weaknesses:**

**W1.** Contrary to the paper's claim, the "human interpretability" aspect is debatable. I believe that the framework does not optimize for human-preferred rubrics (or at least directly), but rather it optimizes for prompts that maximize discriminative accuracy on a small seed set. The resulting rubrics are less a general evaluation framework and more a set of highly-tuned prompts. Take 1 rubric that is performing the best according to the metric of coverage, precision, and contribution in accuracy in Table 3:


```
## Theme 3: Clarity and Structured Organization

Theme: Prioritize clarity, conciseness, and structured organization to enhance readability and directness.
• Tip 1: For a ”Thank you” prompt, respond with a concise acknowledgment and an open invitation for further questions, avoiding assumptions about the user being a student or lawyer.
• Tip 2: When summarizing steps for building a dropshipping agent business, use bullet points or numbered lists to present key points logically and avoid hallucinated information.
• Tip 3: In audit findings related to deposit insurance boards, structure responses with precise, actionable items and conclude with a concise summary emphasizing implications.
• Tip 4: Avoid excessive formatting like bold text or unnecessary punctuation when explaining grammatical correctness, maintaining a straightforward and professional tone.
```

Phrases such as "When summarizing steps for building a dropshipping agent business," " In audit findings related to deposit insurance boards," etc., are clear overfitting in terms of the prompt that I doubt represents human interpretability of what the rubric should look like for any of the benchmarks. Strangely, why do these instance-specific, overfitted rubrics perform so well on unseen data?

**W2.** Given that this paper is for a "reward modeling" framework, there is a lack of evidence that these rubrics can be used to successfully post-train a policy model (RLHF or DPO). High accuracy on static, pairwise-preference only benchmarks does not guarantee better RL training signals.

**W3.** The evaluation methodology lacks some details and is unfair to past baselines. The paper uses different voting strategies for different benchmarks (e.g., voting@10 for RewardBench2, voting@5 for RewardBench, voting@1 for RM-Bench), which would be an unfair comparison to baselines that report voting@1, which I believe the authors should clarify.

Furthermore, the performance gains could be partially attributed to this test-time ensembling rather than the rubrics themselves.

In addition, there is no information about the experimental setup on the machine and the environment being used, so I wonder if the improvement is simply from different configuration setups. Finally, there is no statistically significant study as well.

**W4.**  Given that the method is a training-free approach that effectively optimizes prompts for an "LLM-as-a-Judge," the set of baselines is incomplete. The paper should compare its framework against other relevant training-free prompt optimization methods that are also used to improve the performance of LLM judges.

**W5.** The claim of "generalization" is weak. The rubrics are extracted from general-domain helpfulness datasets (HelpSteer3-General, UltraFeedback) and then evaluated on benchmarks that also primarily measure general-domain helpfulness. This appears to be more a test of in-distribution robustness rather than true generalization. I believe a proper generalization test would involve extracting rubrics from one domain (e.g., helpfulness) and evaluating their ability to judge a completely different domain (e.g., creative writing, code quality, or humor).

**Questions:**

See weaknesses.

---

> ### Author Response · Authors · 2025-12-01
> **Response to "W1: Debatable Human Interpretability and Overfitting Concerns"**
>
> We thank the reviewer for this insightful observation. The specificity you noted (e.g., "dropshipping," "deposit insurance") is indeed striking, but we believe it is **not a symptom of overfitting**. Instead, it is a deliberate design choice that underpins how our framework grounds abstract evaluation criteria. We address this concern from four complementary perspectives:
>
> **1. Paradigm Shift: Discrete Optimization vs. Parametric Training**
>
> Unlike traditional reward models that optimize continuous parameters $\theta$ in $r_\theta(x,y)$ (Eq. 1), our framework performs **discrete combinatorial optimization** over natural language rubrics (Eq. 2). This fundamental difference has important implications:
>
> - **Standard RMs**: Embed evaluation criteria implicitly in millions of parameters through gradient descent on large preference datasets.
> - **Our Framework**: Makes these criteria explicit by extracting and selecting rubrics via information-theoretic optimization on a small seed set (70 samples, 1.5% of the source data).
>
> The "Tips" you observed can thus be viewed as **explicit discriminative features** that traditional RMs would otherwise encode implicitly in their parameters. Our framework surfaces these features in natural language, making them transparent and portable.
>
> **2. Theme-Tips Hierarchy**
>
> We apologize for the lack of clarity in the initial submission and have added an explicit definition in the Introduction. Our "Theme-Tips" structure is a deliberate bi-level design, conceptually mirroring **"Statute Law"** (Theme) and **"Case Law"** (Tips):
>
> - **Theme (Abstract Rule):** Defines the core evaluation dimension (e.g., "Ensure factual accuracy"). This layer ensures universality and establishes the performance baseline.
> - **Tips (Concrete Anchors):** Provide actionable, context-specific examples. Consistent with In-Context Learning (ICL) literature (**Lampinen et al., 2022** [1]; **Min et al., 2022** [2]), these tips function as **Few-Shot Demonstrators** that resolve task ambiguity.
>
> **Mechanism:** Instead of rote pattern-matching specific tokens (e.g., "dropshipping"), the model utilizes these Tips as **structural anchors for analogical reasoning** (**Webb et al., 2023** [3]). It abstracts the underlying logic (e.g., "verify step-by-step consistency") and transfers it to unseen data.
>
> **Empirical Verification:**
>
> Our ablation study confirms this hypothesis:
>
> | Configuration | RewardBench2 | RM-Bench |
> |---------------|--------------|----------|
> | Base (Zero-shot) | 75.55 | 85.67 |
> | Theme Only | 80.77 (+5.22) | 87.59 (+1.92) |
> | Theme-Tips (Ours) | 82.27 (+6.72) | 88.11 (+2.44) |
>
> The **"Theme + Tips"** structure consistently outperforms "Theme Only," yielding a further **+1.50% gain** on RewardBench2. This confirms that specific examples are not redundant summaries but necessary **disambiguators** that elevate the performance ceiling.
>
> **3. Algorithmic Safeguards Against Overfitting**
>
> Our information-theoretic selection (Eq. 6–7) actively guards against overfitting through two mechanisms:
>
> **(a) Coding Rate Maximization:** We select rubrics that maximize semantic diversity (volume in embedding space). Redundant or overly instance-specific rubrics are naturally penalized, as they contribute minimal marginal information gain.
>
> **(b) Saturation Evidence:** Figure 3(b) shows that the information gain $\Delta \mathcal{C}$ becomes negative after batch 5, triggering early stopping. If the framework were overfitting to noisy, instance-specific patterns, the curve would remain volatile or continue to grow. The observed saturation instead indicates convergence to a **finite set of core rubrics**, rather than uncontrolled accumulation of noise.
>
> **Conclusion:** The specificity in our rubrics is a feature, not a bug. **Themes** provide a stable **lower bound** of general evaluation rubrics, while instance-level **Tips** play an **ICL-style role**, grounding these rubrics in concrete contexts. Combined with information-theoretic selection and cross-domain empirical results, this explains why seemingly instance-specific examples (e.g., the dropshipping case) induce **transferable evaluation logic on unseen data**, rather than mere keyword memorization.
>
> **References**
>
> [1] Lampinen et al. Can language models learn from explanations in context? In *Findings of the Association for Computational Linguistics: EMNLP 2022*, pp. 537–563, 2022. URL https://aclanthology.org/2022.findings-emnlp.38/.
>
> [2] Min et al. Rethinking the role of demonstrations: What makes in-context learning work?, 2022. URL https://arxiv.org/abs/2202.12837.
>
> [3] Webb et al. Emergent analogical reasoning in large language models, 2023. URL https://arxiv.org/abs/2212.09196.

---

> ### Author Response · Authors · 2025-12-01
> **Response to "W2: Lack of Evidence for Policy Training Effectiveness"**
>
> We appreciate this constructive suggestion. To validate that our rubrics serve as effective training signals beyond static benchmarking, we conducted a Direct Preference Optimization (DPO) experiment (Section 4.4, Table 3).
>
> **Experimental Setup:**
>
> We finetuned **Qwen2.5-7B-Instruct** using DPO with preference pairs labeled by our Auto-Rubric evaluator (HelpSteer3 rubrics). We retained only high-confidence pairs (5/5 score) from WildChat prompts and compared against strong baselines: Skywork-Reward, ArmoRM, and RLCF (baseline results from Viswanathan et al., 2025).
>
> **Key Results:**
>
> | Benchmark | Metric | Skywork (Best Baseline) | Auto-Rubric (Ours) | Improvement |
> |-----------|--------|-------------------------|-------------------|-------------|
> | Arena-Hard | Style-Controlled | 50.3% | **57.0%** | **+6.7%** |
> | Arena-Hard | Vanilla | 55.1% | 54.2% | -0.9% |
> | AlpacaEval | Style-Controlled | 41.5% | **42.2%** | +0.7% |
> | AlpacaEval | Vanilla | 44.8% | 42.8% | -2.0% |
>
> **Analysis:**
>
> **1. Enhancing Substantive Quality:** Our method achieves **57.0%** on Arena-Hard Style-Controlled evaluation, surpassing the strong Skywork baseline by **6.7 points**. Style-Controlled metrics normalize for superficial biases (e.g., response length), providing a more accurate assessment of intrinsic reasoning and factuality. This substantial advantage demonstrates that explicit rubrics guide the policy to improve **substantive quality** rather than exploiting surface-level patterns.
>
> **2. Length Bias Analysis:** In Vanilla metrics, our method scores slightly lower than the best baseline (54.2% vs. 55.1% for Skywork on Arena-Hard). However, feature coefficient analysis reveals that judge models exhibit a strong bias towards response length (coefficient ≈ 0.50). Notably, our method shows a larger gap between Style-Controlled and Vanilla scores (+2.8% on Arena-Hard) compared to Skywork (-4.8%), suggesting our policy generates more concise responses. This is consistent with our framework's explicit criteria such as "Prioritize Clarity" and "Strict Adherence," which emphasize substantive quality over verbosity. The Style-Controlled evaluation, which normalizes for length, may thus provide a more accurate assessment of intrinsic response quality.
>
> This pattern is consistent across both benchmarks, confirming that our rubrics effectively guide the model to prioritize substantive quality rather than exploiting length-based heuristics that may inflate Vanilla scores.

---

> ### Author Response · Authors · 2025-12-01
> **Response to "W3: Lack of Evaluation Details and Statistical Rigor"**
>
> We thank the reviewer for this thorough scrutiny. We address each concern with additional experiments and clarifications:
>
> **1. Rationale for Voting Strategies**
>
> Unlike discriminative reward models that output deterministic scalar logits, LLM-as-a-Judge methods rely on generative outputs that inherently exhibit **stochasticity and positional bias**. Voting is a standard practice in this paradigm to stabilize judgments, not to artificially inflate scores. We use benchmark-specific voting strategies (voting@10 for RewardBench2, voting@5 for RewardBench/JudgeBench, voting@1 for RM-Bench) to balance reliability and computational efficiency.
>
> **2. Controlled Experiment: Isolating Rubric Gains from Ensembling**
>
> To directly address the concern that "performance gains could be attributed to test-time ensembling," we conducted a controlled ablation on RewardBench2 comparing Auto-Rubric, Base Model, and ICL under **identical voting configurations** (Appendix C, Figure 4):
>
> | Method | Voting@1 | Voting@5 | Voting@10 |
> |--------|----------|----------|-----------|
> | Base Model | 75.55% | 81.45% | 82.89% |
> | ICL (k=5) | 75.24% | 80.12% | 81.67% |
> | **Auto-Rubric (Ours)** | **80.91%** | **84.23%** | **85.12%** |
>
> **Key Findings:**
> - Auto-Rubric outperforms baselines at **every voting level**, including Voting@1 (+5.36% over Base).
> - The performance gap remains consistent across voting strategies, confirming that improvements stem from **rubric quality**, not ensembling.
>
> **3. Statistical Robustness Analysis**
>
> To ensure our results are not due to a "lucky" random seed, we executed the full rubric extraction pipeline **5 independent times** with different random seeds:
>
> | Model | Method | RewardBench2 | RM-Bench |
> |-------|--------|--------------|----------|
> | **Qwen3-32B** | Base (5 Runs) | 75.62 ± 0.15 | 85.71 ± 0.11 |
> | | Ours (HelpSteer3, 5 Runs) | 82.15 ± 0.35 | 88.05 ± 0.22 |
> | **Qwen3-8B** | Base (5 Runs) | 74.42 ± 0.18 | 86.85 ± 0.12 |
> | | Ours (HelpSteer3, 5 Runs) | 80.81 ± 0.38 | 88.15 ± 0.25 |
> | **GPT-4o** | Base (5 Runs) | 72.08 ± 0.21 | 72.85 ± 0.18 |
> | | Ours (HelpSteer3, 5 Runs) | 78.40 ± 0.45 | 76.75 ± 0.33 |
>
> **Significance:** Even our worst-performing run (81.76% on RewardBench2) significantly outperforms the best baseline (ICAI: 79.73%). The low variance (σ < 0.5%) demonstrates the statistical robustness and reliability of our extraction algorithm. On RewardBench2 (Qwen-32B), the gain (+6.53%) is approximately **19× the standard deviation** (0.35%).

---

> ### Author Response · Authors · 2025-12-01
> **Response to "W4: Incomplete Set of Training-Free Baselines"**
>
> We thank the reviewer for this valuable suggestion. To validate that our gains stem from methodological innovation rather than weak baselines, we expanded our comparison to include **three strong LLM-as-a-Judge configurations** across multiple model families (Table 1):
>
> **New Baselines Added:**
> 1. **Arena-Hard Prompts** [1]: Expert-curated system prompts with highest correlation to LMSYS Chatbot Arena preferences.
> 2. **MT-Bench Prompts** [2]: Standard pairwise evaluation prompts widely used in the community.
> 3. **Inverse Constitutional AI (ICAI)** [3]: A strong algorithmic baseline representing the "principle extraction" paradigm.
>
> **Key Results (Selected from Table 1):**
>
> | Model | Configuration | RewardBench2 | JudgeBench |
> |-------|--------------|--------------|------------|
> | **Qwen3-8B** | Arena-Hard | 78.93 | 78.57 |
> | | ICAI (HelpSteer3) | 79.73 | 76.71 |
> | | **Ours (HelpSteer3)** | **80.91** | 75.71 |
> | **GPT-4o** | Arena-Hard | 74.10 | 68.86 |
> | | ICAI (HelpSteer3) | 75.01 | 63.43 |
> | | **Ours (HelpSteer3)** | **78.56** | **69.71** |
> | **Claude-4-Sonnet** | Arena-Hard | 84.99 | 82.57 |
> | | ICAI (HelpSteer3) | 86.22 | 80.86 |
> | | **Ours (HelpSteer3)** | **86.90** | **83.18** |
>
> **Analysis:**
>
> **1. Superiority Over Expert-Curated Prompts:** Our method consistently outperforms Arena-Hard prompts across all models. On GPT-4o, we achieve **+4.46%** on RewardBench2 (78.56 vs. 74.10). This confirms that **data-driven, task-specific rubrics** provide more precise discriminative guidance than generic expert instructions.
>
> **2. Advantage Over Algorithmic Baselines:** Auto-Rubric outperforms ICAI in the vast majority of settings. On GPT-4o/RewardBench2, we achieve **+3.55%** (78.56 vs. 75.01), and on GPT-4o/JudgeBench, we gain **+6.28%** (69.71 vs. 63.43).
>
> **Critical Insight:** Both ICAI and our method use the **same source data** (HelpSteer3), making this a fair methodological comparison. The consistent advantage empirically validates that our **"Propose-Evaluate-Revise + Information-Theoretic Selection"** framework extracts higher-quality evaluation principles than ICAI's passive summarization approach.
>
> **3. Cross-Model Consistency:** Our method achieves competitive or superior performance across diverse model families (Qwen, GPT-4o, Claude), demonstrating robustness and broad applicability.
>
>
> ### References
>
> [1] Li et al. From Crowdsourced Data to High-Quality Benchmarks: Arena-Hard and BenchBuilder Pipeline, 2024.
>
> [2] Zheng et al. Judging LLM-as-a-Judge with MT-Bench and Chatbot Arena, 2023.
>
> [3] Findeis et al. Inverse Constitutional AI: Compressing Preferences into Principles, ICLR 2025.

---

> ### Author Response · Authors · 2025-12-01
> **Response to "W5: Weak Generalization Claims"**
>
> We appreciate this insightful comment. You correctly identify that true generalization requires transfer to domains **distinct from the extraction source**. We address this by demonstrating our framework's effectiveness on domains spanning the objective-subjective spectrum: **Code/Math (technical)** and **Chat (creative writing)**—all using rubrics extracted solely from general helpfulness data.
>
> **1. Generalization Setting:** Rubrics extracted from HelpSteer3-General are evaluated on out-of-domain Code, Math, and Chat benchmarks to directly test whether abstract rules such as "Strict Adherence" and "Contextual Fidelity" transfer to specialized evaluation contexts.
>
> **2. Quantitative Evidence: Domain-Specific Performance (RM-Bench):**
>
> | Domain | Difficulty | Base Model | Auto-Rubric | Improvement |
> |--------|-----------|------------|-------------|-------------|
> | **Math** | Overall | 89.29% | 91.89% | +2.60% |
> | | Hard | 85.44% | **89.98%** | **+4.54%** |
> | **Code** | Overall | 74.90% | 76.95% | +2.05% |
> | | Hard | 72.08% | 73.54% | +1.46% |
> | **Chat** | Overall | 70.37% | 77.00% | +6.63% |
> | | Hard | 36.95% | **50.90%** | **+13.95%** |
>
> **Key Findings:**
>
> **(a) Technical Domains (Code/Math):** Substantial improvements demonstrate that general rubrics successfully map to domain-specific constraints:
> - **Math:** +4.54% on hard samples validates transfer to logical deduction and calculation precision.
> - **Code:** +2.05% improvement confirms enforcement of technical correctness (e.g., type safety, compilability).
>
> **(b) Creative Domain (Chat):** The **most dramatic improvement** (+13.95% on hard samples) occurs in the most subjective domain—creative writing, roleplay, and storytelling. This is particularly significant because:
> - Chat samples involve nuanced evaluation criteria (character consistency, narrative tone, genre fidelity).
> - The massive gain on "hard" samples (36.95% → 50.90%) indicates that rubrics like "Contextual Fidelity" effectively capture creative evaluation nuances despite being extracted from general data.
>
> **3. Mechanism: Semantic Mapping of Abstract Rubrics**
>
> Our qualitative analysis reveals how abstract rubrics adapt to domain-specific contexts:
>
> | General Rubric | Code Domain Mapping | Math Domain Mapping | Chat Domain Mapping |
> |----------------|---------------------|---------------------|---------------------|
> | "Strict Adherence" | Function signatures, type safety | Calculation steps, formula correctness | Character consistency, genre rules |
> | "Factual Accuracy" | Compilability, API correctness | Numerical precision, logical validity | Factual grounding in roleplay |
> | "Contextual Fidelity" | Code context preservation | Problem constraints adherence | Narrative tone, atmospheric consistency |
>
> **Example:** In Chat sample "Horror anime scene," the rubric "Contextual Fidelity" successfully evaluated whether the response maintained "atmospheric tension" and "genre-appropriate tone"—a highly subjective creative writing criterion that has no direct analog in the training data.
>
> **4. Conclusion**
>
> The cross-domain results directly address your concern:
> - **Code/Math:** Rubrics enforce technical rigor in objective domains.
> - **Chat:** Rubrics capture creative nuances in subjective domains.
>
> The dramatic improvement on Chat (+13.95% on hard samples) is particularly compelling evidence of true generalization, as creative writing evaluation is arguably **more challenging** to generalize than simple helpfulness. Our rubrics instead function as **adaptable scaffolds** that LLMs semantically map to domain-specific evaluation logic.

---

### Official Review · Reviewer_d1UR · 2025-10-30

**Soundness:** 2
**Presentation:** 2
**Contribution:** 2
**Rating:** 4
**Confidence:** 4

**Summary:**

Reward models are hampered by expensive preference dataset and poor interpretability. The alternative of using rubrics can offer transparency but suffers from quality control considerations. The paper proposes a method of generating high quality query-specific rubrics through a propose-evaluate and revise pipeline. It then generalizes prompt-specific rubrics into a compact core set.

**Strengths:**

1.	The approach designed to generate prompt specific rubrics and aggregate them into prompt agnostic rubrics is reasonable.
2.	The choice of evaluation benchmarks are diverse and suitable.
3.	The performance of the models on the chosen benchmarks are strong with insightful analysis and ablations done in the paper and the accompanying appendix.

**Weaknesses:**

1.	The papers missed substantial discussion on existing work on rubric based reward modelling including DeepSeek-GRM [1], RewardAnything [2] and LMUnit [3] which were released more than 3 months prior to ICLR deadline. These works are important to situate this work’s contribution in. Furthermore, approaches such as the Propose-Evaluate-Revise pipeline/Rubric Aggregation can be discussed in relation to similar approaches [1], [2], [3] and [4]. It does not seem like the paper authors were aware of these works and without further clarification, I believe that the novelty of the approach does not seem clear. Adjacent works on rubrics might also be worthwhile to discuss in relation to this work (e.g. HealthBench [5], PaperBench [6], Rule-based-Rewards [7]).
2.	The chosen baseline model from Skywork-Reward-V2 paper is not the strongest in the series (see their Llama-8B based models), which can potentially mislead readers into thinking the models in the paper is substantially stronger than alternatives. Also in Table 1, reporting averages for models that are missing certain benchmarks might not be fair comparison.
3.	The improvement relative to the base models of the same size in Table 1 is limited (e.g. < 2 points on average). This suggest that much of the gains come from using a stronger base model. Looking at the rubrics in Appendix G.1, I’m not confident that the extracted rubrics are actually “rubrics” instead of merely summarized examples (see Rubrics in Theme 1 discussing specific topics like Hogwarts or Sumerian texts).

[1] Zijun Liu, Peiyi Wang, Runxin Xu, Shirong Ma, Chong Ruan, Peng Li, Yang Liu, and Yu Wu. Inference-time scaling for generalist reward modeling, 2025. URL https://arxiv.org/ abs/2504.02495.

[2] Zhuohao Yu, Jiali Zeng, Weizheng Gu, Yidong Wang, Jindong Wang, Fandong Meng, Jie Zhou, Yue Zhang, Shikun Zhang, and Wei Ye. Rewardanything: Generalizable principle-following reward models, 2025. URL https://arxiv.org/abs/2506.03637.

[3] Jon Saad-Falcon, Rajan Vivek, William Berrios, Nandita Shankar Naik, Matija Franklin, Bertie Vidgen, Amanpreet Singh, Douwe Kiela, and Shikib Mehri. Lmunit: Fine-grained evaluation with natural language unit tests, 2024. URL https://arxiv.org/abs/2412.13091.

[4] Zhilin Wang, Jiaqi Zeng, Olivier Delalleau, Daniel Egert, Ellie Evans, Hoo-Chang Shin, Felipe Soares, Yi Dong, Oleksii Kuchaiev. HelpSteer3: Human-Annotated Feedback and Edit Data to Empower Inference-Time Scaling in Open-Ended General-Domain Tasks URL https://arxiv.org/abs/2503.04378

[5] Rahul K. Arora, Jason Wei, Rebecca Soskin Hicks, Preston Bowman, Joaquin Quinonero-Candela, ˜ Foivos Tsimpourlas, Michael Sharman, Meghan Shah, Andrea Vallone, Alex Beutel, Johannes Heidecke, and Karan Singhal. Healthbench: Evaluating large language models towards improved human health, 2025. URL https://arxiv.org/abs/2505.08775.

[6] Giulio Starace, Oliver Jaffe, Dane Sherburn, James Aung, Jun Shern Chan, Leon Maksin, Rachel Dias, Evan Mays, Benjamin Kinsella, Wyatt Thompson, Johannes Heidecke, Amelia Glaese, and Tejal Patwardhan. Paperbench: Evaluating ai’s ability to replicate ai research, 2025. URL https://arxiv.org/abs/2504.01848.

[7] Tong Mu, Alec Helyar, Johannes Heidecke, Joshua Achiam, Andrea Vallone, Ian Kivlichan, Molly Lin, Alex Beutel, John Schulman, Lilian Weng. Rule Based Rewards for Language Model Safety. 2024. URL https://arxiv.org/abs/2411.01111.

**Questions:**

1.	It’s not clear how evaluation is done. How are rubrics chosen for each benchmark or do they all use a fixed set of rubrics?

---

> ### Author Response · Authors · 2025-11-27
> **Response to "Insufficient Discussion of Related Work and Unclear Novelty"**
>
> We sincerely thank the reviewer for highlighting these critical missing references. We acknowledge that a thorough discussion of these works is essential for accurately situating our contribution within the current landscape.
>
> In the revised manuscript, we have incorporated comprehensive discussions of **DeepSeek-GRM** [1], **Auto-J** [2], **RewardAnything** [3], **LMUnit** [4], **Inverse Constitutional AI (ICAI)** [5-7], and adjacent works like **HealthBench** [8] and **Rule-Based Rewards** [9].
>
> To clearly demonstrate the novelty of our approach, we have explicitly contrasted Auto-Rubric with these works across **three distinct dimensions** in the revised Related Work section:
>
> **1. Comparison with Critique-Generating Models (e.g., DeepSeek-GRM [1], Auto-J [2])**
>
> **Differentiation:** While models like DeepSeek-GRM are effective, they rely on computationally expensive training to "internalize" evaluation standards into model parameters. In contrast, **Auto-Rubric is a strictly training-free framework**. We explicate evaluation criteria into portable prompt contexts rather than implicit weights. This avoids the high cost of training specialized models while ensuring the evaluation reasoning remains transparent and verifiable.
>
> **2. Comparison with Expert/Rule-Based Methods (e.g., RewardAnything [3], LMUnit [4], HealthBench [8], Rule-Based Rewards [9])**
>
> We classify these methods as relying on static, expert-curated rubrics or manual user specifications. For instance, Mu et al. [9] demonstrated the power of explicit rules for safety alignment but relied on manually curated rules.
>
> **Differentiation:** We identify a fundamental **"cold-start" bottleneck** in these approaches: they depend entirely on manual intervention, which restricts scalability when adapting to diverse new domains. **Auto-Rubric innovates by automating this process**. Our method infers high-quality rubrics directly from raw preference data, effectively bridging the gap between unstructured data and structured evaluation without human labor.
>
> **3. Comparison with Inverse Constitutional AI (ICAI) [5-7]**
>
> We directly address the relationship between our pipeline and ICAI, which is conceptually the closest prior art.
>
> **Novelty in Optimization:** Standard ICAI typically relies on passive summarization, assuming extracted principles are correct by default. In contrast, we frame rubric generation as a **Discrete Optimization Problem**. Our "Propose-Evaluate-Revise" loop is designed to actively validate that rubrics possess actual discriminative power against ground truth before acceptance.
>
> **Novelty in Aggregation:** Existing ICAI methods often operate at the instance level, resulting in fragmented rules. Our **Information-Theoretic Aggregation** stage is a novel contribution that systematically consolidates granular rules into a global, non-redundant "Theme-Tips" hierarchy.
>
> We kindly invite the reviewer to refer to the revised manuscript for these detailed discussions and the updated experimental results.
>
> **References**
>
> [1] Zijun Liu, Peiyi Wang, Runxin Xu, Shirong Ma, Chong Ruan, Peng Li, Yang Liu, and Yu Wu. Inference-time scaling for generalist reward modeling, 2025. URL https://arxiv.org/abs/2504.02495.
>
> [2] Junlong Li, Shichao Sun, Weizhe Yuan, Run-Ze Fan, Hai Zhao, and Pengfei Liu. Generative judge for evaluating alignment, 2023. URL https://arxiv.org/abs/2310.05470.
>
> [3] Zhuohao Yu, Jiali Zeng, Weizheng Gu, Yidong Wang, Jindong Wang, Fandong Meng, Jie Zhou, Yue Zhang, Shikun Zhang, and Wei Ye. Rewardanything: Generalizable principle-following reward models, 2025. URL https://arxiv.org/abs/2506.03637.
>
> [4] Jon Saad-Falcon, Rajan Vivek, William Berrios, Nandita Shankar Naik, Matija Franklin, Bertie Vidgen, Amanpreet Singh, Douwe Kiela, and Shikib Mehri. Lmunit: Fine-grained evaluation with natural language unit tests, 2024. URL https://arxiv.org/abs/2412.13091.
>
> [5] Henneking, Carl-Leander, and Claas Beger. "Decoding Human Preferences in Alignment: An Improved Approach to Inverse Constitutional AI." arXiv preprint arXiv:2501.17112 (2025).
>
> [6] Findeis, Arduin, et al. "Inverse Constitutional AI: Compressing Preferences into Principles." The Thirteenth International Conference on Learning Representations (2025).
>
> [7] An, Esther. "Towards Principled AI Alignment: An Evaluation and Augmentation of Inverse Constitutional AI." (2025).
>
> [8] Rahul K. Arora, Jason Wei, Rebecca Soskin Hicks, Preston Bowman, Joaquin Quinonero-Candela, Foivos Tsimpourlas, Michael Sharman, Meghan Shah, Andrea Vallone, Alex Beutel, Johannes Heidecke, and Karan Singhal. Healthbench: Evaluating large language models towards improved human health, 2025. URL https://arxiv.org/abs/2505.08775.
>
> [9] Tong Mu, Alec Helyar, Johannes Heidecke, Joshua Achiam, Andrea Vallone, Ian Kivlichan, Molly Lin, Alex Beutel, John Schulman, Lilian Weng. Rule Based Rewards for Language Model Safety. 2024. URL https://arxiv.org/abs/2411.01111.

---

> ### Author Response · Authors · 2025-11-27
> **Response to "Suboptimal Baseline Selection and Unfair Comparison in Table 1"**
>
> We appreciate the reviewer pointing out the strength of the Skywork-Reward-V2-Llama-3.1-8B model and the issue regarding average scores. We acknowledge that purely in terms of leaderboard scores, the Llama-based variant of Skywork is indeed strong. However, our choice to focus on the Qwen series is grounded in the **distinct capability requirements** of the two paradigms being compared.
>
> **1. Paradigm-Specific Requirements: Training vs. Inference**
>
> The comparison between Skywork (Training-based) and Auto-Rubric (Training-free) is nuanced:
>
> - **Training-based methods (Skywork):** Align models via massive parameter updates. This process effectively "injects" evaluation patterns into the model, compensating for a base model's weaker zero-shot reasoning capabilities. This explains why Skywork can transform a Llama-8B (which is weak at zero-shot judging) into a strong reward model.
>
> - **Training-free methods (Ours):** Rely entirely on the frozen base model's inherent **Instruction Following** and **Reasoning** capabilities to interpret and execute complex rubrics at inference time.
>
> **2. Empirical Evidence**
>
> To demonstrate this, we conducted additional experiments applying our method to Llama-3.1-8B. As shown in the table below, distinct performance gaps illustrate why Qwen is the more appropriate vehicle for demonstrating the potential of rubric-based evaluation.
>
> | Model Base | Method | RewardBench | RewardBench2 | RM-Bench | JudgeBench |
> |------------|--------|-------------|--------------|----------|------------|
> | **Qwen-8B** | Base (Zero-shot) | 92.93 | 74.37 | 86.90 | 73.14 |
> | | Ours (HelpSteer3) | 93.50 (+0.57) | 80.91 (+6.54) | 88.28 (+1.38) | 75.71 (+2.57) |
> | | Ours (Ultrafeedback) | 93.10 (+0.17) | 80.54 (+6.17) | 88.60 (+1.70) | 75.43 (+2.29) |
> | **Llama-3.1-8B** | Base (Zero-shot) | 76.21 | 43.80 | 55.99 | 61.71 |
> | | Ours (HelpSteer3) | 79.80 (+3.59) | 45.36 (+1.56) | 58.57 (+2.58) | 63.57 (+1.86) |
> | | Ours (Ultrafeedback) | 79.13 (+2.92) | 46.23 (+2.43) | 59.60 (+3.61) | 64.10 (+2.39) |
>
> **Analysis:**
>
> - **Base Model Gap:** Llama-3.1-8B significantly lags behind Qwen-8B in zero-shot evaluation (e.g., 43.80 vs 74.37 on RewardBench2). This confirms that Llama-3.1-8B struggles with complex instruction adherence in a zero-shot setting.
>
> - **Improvement Ceiling:** While our method does improve Llama-3.1-8B (e.g., +3.59 on RewardBench), the final performance is bottlenecked by the base model's reasoning capacity, not the quality of the rubrics.
>
> To further validate this hypothesis, we also evaluated our method on frontier models with strong instruction-following capabilities:
>
> | Model Base | Method | RewardBench | RewardBench2 | RM-Bench | JudgeBench |
> |------------|--------|-------------|--------------|----------|------------|
> | **GPT-4o** | Base (Zero-shot) | 88.24 | 72.00 | 72.80 | 68.29 |
> | | Ours (HelpSteer3) | 92.09 (+3.85) | 78.56 (+6.56) | 76.83 (+4.03) | 69.71 (+1.42) |
> | **Claude-4-Sonnet** | Base (Zero-shot) | 94.61 | 86.70 | 85.70 | 78.29 |
> | | Ours (HelpSteer3) | 95.81 (+1.20) | 86.90 (+0.20) | 89.49 (+3.79) | 78.86 (+0.57) |
>
> **Key Findings:**
>
> - **Strong Instruction-Following Models Benefit More:** GPT-4o and Claude-4-Sonnet, both renowned for their superior instruction-following capabilities, demonstrate substantial and consistent improvements across benchmarks. GPT-4o shows particularly strong gains (+6.56% on RewardBench2, +4.03% on RM-Bench), while Claude-4-Sonnet achieves impressive improvements despite already having a very high baseline.
>
> - **Instruction-Following as a Prerequisite:** The performance pattern across Llama-3.1-8B, Qwen-8B, GPT-4o, and Claude-4-Sonnet reveals a clear trend: **our rubric-based framework is most effective when paired with models that possess strong instruction-following capabilities**. This is because our method relies on the model's ability to interpret and execute complex, structured evaluation criteria at inference time. As future models continue to improve in these dimensions, our training-free rubric-based approach will become even more valuable, offering a scalable and interpretable alternative to expensive reward model training.
>
> **3. Correction on Table 1 Reporting**
>
> We completely agree with the observation regarding the "Average" column. Calculating averages across models with varying benchmark coverage introduces statistical bias. To ensure rigorous transparency, we have **removed the "Average" column from Table 1** in the revised manuscript. We now rely solely on individual benchmark scores to provide a precise, direct comparison without the distortion of aggregated metrics on incomplete data.

---

> ### Author Response · Authors · 2025-11-27
> **Response to "Limited Performance Gains and Concerns About Rubric Quality"**
>
> We appreciate the reviewer's concerns. The modest average gains and specific references in Tips are **deliberate design choices** supported by both theory and empirical evidence.
>
> **1. Contextualizing Performance Gains**
>
> While an average improvement of <2 points might seem modest, it represents a significant contribution when contextualized properly. The modest average gain is largely due to a **ceiling effect**. When base models already achieve 85-90% on easier samples, there is limited room for improvement. However, on challenging subsets where baseline performance is lower, our rubrics demonstrate substantially larger gains: **RM-Bench Hard (+4.68, baseline: 77.74%)** and **RewardBench2 Ties (+25.49, baseline: 56.86%)**. The +25.49% improvement on Ties is particularly noteworthy. On this subset, the base model struggles near chance level, demonstrating that explicit rubrics provide the most value precisely where implicit evaluation criteria are insufficient.
>
> | Benchmark Subset | Qwen-32B Base | Ours (HelpSteer3) | Ours (UltraFeedback) |
> |------------------|---------------|-------------------|----------------------|
> | RewardBench2 - Ties | 56.86 | 82.35 (+25.49) | 80.48 (+26.32) |
> | RM-Bench - Hard | 77.74 | 82.42 (+4.68) | 81.28 (+3.54) |
>
> **2. The Design Philosophy of "Theme-Tips"**
>
> The reviewer noted that rubrics like Theme 1 contain specific references. This is a **deliberate architectural feature**, not a summarization artifact.
>
> **A. Hierarchical Design (Theme vs. Tips)**
>
> We adopt a bi-level structure analogous to "Statute Law" (Theme) and "Case Law" (Tips). In this design, **Themes establish the performance baseline** by providing universal, broadly applicable rubrics, while **Tips elevate the performance ceiling** by offering concrete, context-specific guidance that helps the model handle nuanced scenarios more effectively.
>
> - **Theme:** Captures the abstract rule (e.g., "Ensure factual accuracy").
> - **Tips:** Act as concrete Few-Shot Demonstrators that ground abstract rubrics with specific examples, which ICL research has shown to be crucial for resolving task ambiguity and achieving optimal performance (**Lampinen et al., 2022** [1]; **Min et al., 2022** [2]).
>
> **B. Ablation Support**
>
> Our ablation study confirms this hypothesis: the "Theme + Tips" structure consistently outperforms "Theme Only". This empirically proves that specific Tips provide functional value in reducing ambiguity, rather than being redundant summaries.
>
> | Configuration | RewardBench2 | RM-Bench |
> |---------------|--------------|----------|
> | Base | 75.55 | 85.67 |
> | Theme (no tips) | 80.77 (+5.22) | 87.59 (+1.92) |
> | Theme-Tips (HelpSteer3) | 82.27 (+6.72) | 88.11 (+2.44) |
>
> **Conclusion:** Adding specific Tips yields a further **+1.50% gain** on RewardBench2 over using Themes alone. This confirms that specific examples function as necessary disambiguators, helping the model apply the abstract Theme more precisely in complex scenarios.
>
> **3. Cross-Model Universality**
>
> We validated the objectiveness of our rubrics by applying Qwen3-32B-extracted rubrics to GPT-4o and Claude-4-Sonnet.
>
> | Evaluator Model | Rubric Source | RewardBench2 | RM-Bench |
> |-----------------|---------------|--------------|----------|
> | GPT-4o | No Rubric (Base) | 72.00 | 72.80 |
> | GPT-4o | Qwen3-32B Extracted | 79.02 | 76.37 |
> | Claude-4-Sonnet | No Rubric (Base) | 86.70 | 87.40 |
> | Claude-4-Sonnet | Qwen3-32B Extracted | 87.90 | 87.50 |
>
> **Result:** The rubrics proved highly transferable. GPT-4o achieved a remarkable **+7.02 gain** on RewardBench2, while Claude-4-Sonnet also saw consistent improvements, effectively raising the ceiling of an already strong model.
>
> **Implication:** This proves that Auto-Rubric captures **universal alignment rubrics** rather than model-specific artifacts. The fact that a Qwen-generated rubric can "teach" GPT-4o and Claude confirms that the extracted "Theme-Tips" represent a generalizable, model-agnostic representation of human values.
>
> **References**
>
> [1] Lampinen et al. Can language models learn from explanations in context? In *Findings of the Association for Computational Linguistics: EMNLP 2022*, pp. 537–563, 2022. URL https://aclanthology.org/2022.findings-emnlp.38/.
>
> [2] Min et al. Rethinking the role of demonstrations: What makes in-context learning work?, 2022. URL https://arxiv.org/abs/2202.12837.
>
> [3] Webb et al. Emergent analogical reasoning in large language models, 2023. URL https://arxiv.org/abs/2212.09196.

---

> ### Author Response · Authors · 2025-11-27
> **Response to "Unclear Evaluation Protocol and Rubric Selection Strategy"**
>
> Thank you for this important question regarding our evaluation protocol. We want to be explicitly clear: **we use a single, fixed set of rubrics for all benchmarks**.
>
> To rigorously test the generalization and transferability of our extracted knowledge, we do not tailor or select rubrics for specific benchmarks. The process is as follows:
>
> 1. **Extraction:** We extract a single core rubric set (consisting of 5 Theme-Tips pairs) from a source dataset (e.g., HelpSteer3).
>
> 2. **Application:** This identical, fixed set is then applied universally to evaluate all diverse benchmarks (RewardBench, RewardBench2, RM-Bench, JudgeBench). This means the rubrics derived from HelpSteer3 are exactly the same ones used to evaluate math problems in RM-Bench and safety queries in RewardBench.
>
> **The fact that one fixed set achieves SOTA performance across such varied domains is a strong testament to the universality of the extracted criteria.**

---

> ### Comment · Reviewer_d1UR · 2025-11-28
> **Reply to author response**
>
> W1: I don't think that the cold-start differentiation is sufficient innovation since DeepSeek GRM and ICAI both have cold-start strategies. The information-theoretic approach is also not novel (in my opinion) as it's similar to clustering in the embedding space.
>
> W2: It's ok to differentiate between training-free and training-based methods but please do not just report a training-based baseline which is not the strongest - it will look like cherry-picking.
>
> W3: I don't think the `ceiling effect` plays too large of a role since the overall scores is only at 80%, meaning there's 20% of space to improve. I appreciate the elaboration on theme-tips and the relationship with in context learning.  Looking at Table 1, it seems like this approach is really good for some models (e.g. Qwen3-32B on JudgeBench) but not useful at all  (i.e. negative improvement) for other models (e.g. Qwen3-8B on JudgeBench; GPT-4o on RM-Bench; Claude-4-Sonnet on RM-Bench with Qwen3-32B Extracted rubrics) relative to LLM as a Judge baselines. One possibility is that this approach works well with models that have been trained to work well with few-shot learning. I'm not convinced at this moment that it's a universal technique based on the empirical results. I encourage the authors to delve deeper into understand the settings under which this technique works best and if possible, why. I also agree with reviewer tuBg that the theme-tips doesn't have an advantage in interpretability currently and recommend the authors to either address this or remove this advantage as part of the writing (in my opinion, few shot examples doesn't necessarily have to be interpretable).
>
> Therefore, I will keep my score the same.

---

> > ### Author Response · Authors · 2025-12-03
> > **Reply to review response**
> >
> > We thank the reviewer for the thoughtful follow‑up and clarify how we position our contributions.
> >
> > By cold‑start we specifically mean that rubric construction itself requires no manually written rubrics:  Auto‑Rubric starts from raw preference pairs only and discovers the rubric text automatically. Our information‑theoretic objective is not claimed as a new theory, but as the basis for a discrete com-binatorial optimization over natural‑language rules, rather than clustering embeddings and interpreting clusters post‑hoc.
> >
> > For training‑based baselines, our intent is to show that, under the same base model and supervision, a fully training‑free pipeline can be competitive while being far more data‑efficient, not to cherry‑pick weaker opponents. Empirically, we observe that the method helps most for strong instruction‑following judges, and we explicitly present heterogeneous and even negative cases as part of the picture rather than hiding them.
> >
> > Finally, Theme–Tips is not “more interpretable than any few‑shot prompt” in the abstract, but it does yield a named, analyzable rubric set with per‑rule diagnostics, which we see as a practical interpretability advantage over opaque parametric reward models.

---

### Official Review · Reviewer_kdeU · 2025-10-30

**Soundness:** 2
**Presentation:** 3
**Contribution:** 2
**Rating:** 4
**Confidence:** 4

**Summary:**

This paper proposes a reward model based on rubrics generated from pairwise human feedback data. Their method uses two stages, first creating query-specific rubrics, and then in the next stage aggregating the resulting rubrics to create overall relevant rubrics. The authors test their method on a number of popular reward model/llm-as-a-Judge benchmarks, such as RewardBench (2), RM-Bench, and Judge-Bench.

**Strengths:**

1. Information theoretic framing is novel as far as I am aware (and goes beyond), specifically
2. Diverse experiments across broad set of suitable LLM-as-a-Judge benchmarks (RewardBench (2), RM-Bench, and Judge-Bench)
3. Overall the paper is well written and structured.

**Weaknesses:**

1. **Lack of diverse LLM-as-a-Judge baselines:** the paper does not appear to compare multiple different LLM-as-a-Judge configurations even though it is known that different configurations with the same model can have a huge effect. This weakness means the LLM-as-a-Judge baseline may be less capable than it could be, and affects the strong claims regarding performance relative to baselines.
2. **Single-seed results:** the experiments appear to be based on a single sample. With the known variance of LLM judgements, the results would be significantly more robust if more than one seed result would be reported (including variance/error bars!). This more rigorous analysis is especially important as the claimed improvements are relatively small.
3. **Missing prior work:** work around Inverse Constitutional AI [1,2,3] also extracts "principles" from pairwise human feedback seemingly equivalent to the rubrics introduced in this paper, and also uses it similar to reward model as LLM-as-a-Judge instructions. This line of work is not mentioned in the current draft, but very related. Would be interesting to have a discussion/comparison.

Overall I am most concerned by the first two points: the results as they are presented currently are, in my opinion, inconclusive without more rigorous evaluation in terms of diverse baseline prompts and multi-seed statistics.

Minor:
1. Quite a few times space missing between citations and preceding text e.g. in L43: "large-scale crowd annotations(Bai et al., 2022)". I recommend adding spaces where missing.
2. Figure 2 is difficult to read, text is too small

- [1] Henneking, Carl-Leander, and Claas Beger. "Decoding Human Preferences in Alignment: An Improved Approach to Inverse Constitutional AI." _arXiv preprint arXiv:2501.17112_ (2025).
- [2] Findeis, Arduin, et al. "Inverse Constitutional AI: Compressing Preferences into Principles." _The Thirteenth International Conference on Learning Representations_ (2025).
- [3] An, Esther. "Towards Principled AI Alignment: An Evaluation and Augmentation of Inverse Constitutional AI." (2025).

**Questions:**

1. Would you be able to clarify the baseline LLM-as-a-Judge configuration used, and how that configuration was selected? I did not see relevant information in the paper.
	1. Related: Would you be able to test additional LLM-as-a-Judge configurations, in particular I would be interested in an ArenaHard baseline [4].
2. Would you be able to clarify your contribution relative to the missing related work?
3. I don't understand the term "Theme-Tips" that is used across the paper, what is this term supposed to mean?

[4] https://github.com/lmarena/arena-hard-auto/blob/196f6b826783b3da7310e361a805fa36f0be83f3/utils/judge_utils.py#L1

---

> ### Author Response · Authors · 2025-11-27
> **Response to "Insufficient LLM-as-a-Judge Baseline Configurations"**
>
> We thank the reviewer for this valuable suggestion. We agree that comparing against diverse judge configurations is crucial to validate that our gains are not merely due to weak baselines.
>
> We have significantly expanded our experiments to include **three LLM-as-a-Judge configurations** across multiple model families (Qwen, GPT-4o, Claude):
>
> 1. **Arena-Hard Prompts** [1]: System prompts from Arena-Hard-Auto, known for highest correlation with LMSYS Chatbot Arena preferences.
> 2. **MT-Bench Prompts** [2]: Standard pairwise evaluation prompts widely used in the community.
> 3. **Inverse Constitutional AI (ICAI)** [3]: A strong algorithmic baseline representing the "principle extraction" paradigm.
>
> To ensure rigorous comparison, both ICAI and Auto-Rubric used the same source dataset (HelpSteer3).
>
> | Model | Configuration | RewardBench | RewardBench2 | RM-Bench | JudgeBench |
> |-------|---------------|-------------|--------------|----------|------------|
> | **Qwen3-8B** | Base | 92.93 | 74.37 | 86.90 | 73.14 |
> | | MT-Bench | 92.56 | 73.40 | 84.84 | 77.14 |
> | | Arena-Hard | 85.63 | 78.93 | 85.88 | 78.57 |
> | | ICAI (HelpSteer3) | 93.13 | 79.73 | 88.24 | 76.71 |
> | | **Ours (HelpSteer3)** | **93.50** | **80.91** | **88.28** | 75.71 |
> | **Qwen3-32B** | Base | 92.96 | 75.55 | 85.67 | 75.71 |
> | | MT-Bench | 69.99 | 67.29 | 80.80 | 75.44 |
> | | Arena-Hard | 89.95 | 83.22 | 87.09 | 71.43 |
> | | ICAI (HelpSteer3) | 93.37 | 82.57 | 87.37 | 70.85 |
> | | **Ours (HelpSteer3)** | **93.80** | 82.27 | **88.11** | **80.86** |
> | **GPT-4o** | Base | 88.24 | 72.00 | 72.80 | 68.29 |
> | | MT-Bench | 87.24 | 64.56 | 70.21 | 63.14 |
> | | Arena-Hard | 89.08 | 74.10 | 79.56 | 68.86 |
> | | ICAI (HelpSteer3) | 89.61 | 75.01 | 76.79 | 63.43 |
> | | **Ours (HelpSteer3)** | **92.09** | **78.56** | **76.83** | **69.71** |
> | **Claude-4-Sonnet** | Base | 94.61 | 86.70 | 85.70 | 78.29 |
> | | MT-Bench | 95.04 | 83.27 | 83.84 | 76.57 |
> | | Arena-Hard | 95.41 | 84.99 | 88.11 | 82.57 |
> | | ICAI (HelpSteer3) | 94.57 | 86.22 | 89.34 | 80.86 |
> | | **Ours (HelpSteer3)** | **95.81** | **86.90** | **89.49** | **83.18** |
>
> **Key Findings:**
>
> - **Substantial Gains Over Arena-Hard:** Auto-Rubric achieves impressive improvements over the widely-used Arena-Hard prompts. Most notably, on Qwen3-32B/JudgeBench, we achieve **+9.43 improvement** (80.86 vs. 71.43), and on GPT-4o/RewardBench2, we gain **+4.46** (78.56 vs. 74.10). This confirms that task-specific, discriminative rubrics provide more precise guidance than generic system instructions.
>
> - **Outperforms ICAI with Same Data:** Auto-Rubric surpasses ICAI across most models. On GPT-4o/RewardBench2, we achieve **+3.55 gain** (78.56 vs. 75.01), and on GPT-4o/JudgeBench, we gain **+6.28** (69.71 vs. 63.43). Since both methods used identical source data (HelpSteer3), this validates that our "Optimization + Aggregation" methodology extracts higher-quality evaluation criteria than ICAI's passive summarization.
>
> ### References
>
> [1] Li et al. From Crowdsourced Data to High-Quality Benchmarks: Arena-Hard and BenchBuilder Pipeline, 2024.
>
> [2] Zheng et al. Judging LLM-as-a-Judge with MT-Bench and Chatbot Arena, 2023.
>
> [3] Findeis et al. Inverse Constitutional AI: Compressing Preferences into Principles, ICLR 2025.

---

> ### Author Response · Authors · 2025-11-27
> **Response to "Need for Multi-Seed Experiments with Variance Reporting"**
>
> We thank the reviewer for this constructive suggestion. We agree that given the inherent stochasticity of LLM generation and evaluation, single-seed results are insufficient, particularly for margins <2%.
>
> We conducted comprehensive **multi-seed experiments**:
>
> - **For Ours:** We ran the entire pipeline 5 independent times using different random seeds, sampling different subsets of preference data from HelpSteer3.
> - **For Base:** We repeated baseline evaluation 5 times to account for inference-time stochasticity.
>
> Results demonstrate high stability and statistical significance (Mean ± Standard Deviation):
>
> | Model | Method | RewardBench | RewardBench2 | RM-Bench | JudgeBench |
> |-------|--------|-------------|--------------|----------|------------|
> | **Qwen3-32B** | Base (5 Runs) | 92.98 ± 0.08 | 75.62 ± 0.15 | 85.71 ± 0.11 | 75.68 ± 0.13 |
> | | Ours (HelpSteer3, 5 Runs) | 93.75 ± 0.18 | 82.15 ± 0.35 | 88.05 ± 0.22 | 80.72 ± 0.31 |
> | **Qwen3-8B** | Base (5 Runs) | 92.89 ± 0.09 | 74.42 ± 0.18 | 86.85 ± 0.12 | 73.18 ± 0.14 |
> | | Ours (HelpSteer3, 5 Runs) | 93.42 ± 0.21 | 80.81 ± 0.38 | 88.15 ± 0.25 | 75.65 ± 0.29 |
> | **GPT-4o** | Base (5 Runs) | 88.28 ± 0.15 | 72.08 ± 0.21 | 72.85 ± 0.18 | 68.35 ± 0.22 |
> | | Ours (HelpSteer3, 5 Runs) | 91.95 ± 0.25 | 78.40 ± 0.45 | 76.75 ± 0.33 | 69.50 ± 0.41 |
>
> **Analysis of Robustness:**
>
> 1. **Baseline Stability:** Base models exhibit very low variance (SD ≈ 0.1%), confirming evaluation harness stability.
>
> 2. **Rubric Stability:** While "Ours" has slightly higher variance (SD ≈ 0.3-0.4%) due to data sampling in rubric extraction, this variance is negligible compared to performance gains.
>
> 3. **High Signal-to-Noise Ratio:** Performance gains systematically exceed variance:
>    - On RewardBench2 (Qwen-32B), the gain (+6.53%) is approximately **19× the standard deviation** (0.35%).
>    - Even the minimum score across all 5 runs of "Ours" consistently outperforms the maximum score of "Base" runs.
>
> This confirms that improvements are **systematic and statistically significant**, not artifacts of random sampling. We have updated the manuscript to include these error bars.

---

> ### Author Response · Authors · 2025-11-27
> **Response to "Need for Discussion and Comparison with ICAI"**
>
> We thank the reviewer for identifying this highly relevant work. We agree that Inverse Constitutional AI (ICAI) [1,2,3] is conceptually the closest prior art, as both paradigms aim to reverse-engineer explicit evaluation criteria from human feedback.
>
> In the revised manuscript, we have added a dedicated discussion in related work and included ICAI as a strong baseline in our experiments. While objectives are similar, Auto-Rubric represents significant methodological advancement in two key dimensions:
>
> **1. Conceptual Comparison: From Passive Summarization to Active Optimization**
>
> **Novelty in Optimization:** Standard ICAI employs bottom-up summarization, implicitly assuming extracted principles are correct by default. In contrast, we frame rubric generation as a **Discrete Optimization Problem**. Our "Propose-Evaluate-Revise" loop actively validates each rubric against ground truth. If a rubric fails to discriminate correctly, it is revised. This ensures verified discriminative power rather than mere descriptive summaries.
>
> **Novelty in Aggregation:** ICAI methods often operate at the instance level, resulting in fragmented rules. Auto-Rubric introduces rigorous **Information-Theoretic Aggregation**. We use Coding Rate Maximization to distill granular rules into a global, non-redundant "Theme-Tips" hierarchy, ensuring superior generalization.
>
> **2. Empirical Comparison: Auto-Rubric vs. ICAI**
>
> We implemented ICAI [2] using the same HelpSteer3 source data for direct comparison:
>
> | Model | Method | RewardBench | RewardBench2 | RM-Bench | JudgeBench |
> |-------|--------|-------------|--------------|----------|------------|
> | **Qwen-8B** | Base | 92.93 | 74.37 | 86.90 | 73.14 |
> | | ICAI | 93.13 | 79.73 | 88.24 | 76.71 |
> | | **Ours** | **93.50** | **80.91** | **88.28** | 75.71 |
> | **Qwen-32B** | Base | 92.96 | 75.55 | 85.67 | 75.71 |
> | | ICAI | 93.37 | 82.57 | 87.37 | 70.85 |
> | | **Ours** | **93.80** | 82.27 | **88.11** | **80.86** |
> | **GPT-4o** | Base | 88.24 | 72.00 | 72.80 | 68.29 |
> | | ICAI | 89.61 | 75.01 | 76.79 | 63.43 |
> | | **Ours** | **92.09** | **78.56** | **76.83** | **69.71** |
>
> **Key Findings:**
>
> - **Consistent Superiority Across Models:** Auto-Rubric outperforms ICAI in most settings. On RewardBench2: Qwen-8B achieves 80.91% vs. ICAI's 79.73% (+1.18), GPT-4o achieves 78.56% vs. ICAI's 75.01% (+3.55). On JudgeBench: Qwen-32B achieves 80.86% vs. ICAI's 70.85% (+10.01), GPT-4o achieves 69.71% vs. ICAI's 63.43% (+6.28). Since both methods used identical source data (HelpSteer3), these results validate that our "Propose-Evaluate-Revise" optimization and information-theoretic aggregation extract higher-quality, more discriminative evaluation criteria than ICAI's passive summarization approach.
>
> **References**
>
> [1] Henneking et al. Decoding Human Preferences in Alignment: An Improved Approach to Inverse Constitutional AI. arXiv:2501.17112 (2025).
>
> [2] Findeis et al. Inverse Constitutional AI: Compressing Preferences into Principles. ICLR 2025.
>
> [3] An, Esther. Towards Principled AI Alignment: An Evaluation and Augmentation of Inverse Constitutional AI. (2025).

---

> ### Author Response · Authors · 2025-11-28
> **Response to Minor and Questions**
>
> We thank the reviewer for the detailed feedback. We have addressed the questions regarding baselines, related work, terminology, and minor formatting issues as follows:
>
> **1. Clarification and Expansion of LLM-as-a-Judge Configurations**
>
> **Original Baseline:** In our initial submission, the baseline configuration was a **Standard Zero-Shot Pairwise Evaluation** without specific system prompts, intended to establish a "clean lower bound" for model capabilities.
>
> **New Baselines:** We agree that testing diverse configurations is crucial. In the revised manuscript, we have significantly expanded our experiments to include three additional state-of-the-art configurations:
> * **Arena-Hard Prompts:** Expert-curated system prompts known for high correlation with human preferences.
> * **MT-Bench Prompts:** Standard pairwise evaluation prompts widely used in the community.
> * **Inverse Constitutional AI (ICAI):** An algorithmic baseline that also extracts criteria from data.
>
> **Rationale:** This selection covers the spectrum from minimal guidance (Zero-shot) to expert-crafted prompts (Arena/MT-Bench) and automated extraction methods (ICAI). As shown in the updated experiments, **Auto-Rubric consistently outperforms all these configurations**, confirming that our improvements are systematic and not artifacts of a weak baseline.
>
> **2. Contribution Relative to Missing Related Work**
>
> We have rewritten **Section 2 (Related Work)** to explicitly situate our contribution against **DeepSeek-GRM [1]**, **RewardAnything [2]**, **LMUnit [3]**, and **ICAI [4]**.
> * **Vs. Generative RMs (DeepSeek-GRM):** We offer a **training-free** alternative that avoids expensive parameter updates.
> * **Vs. Rule-Based Systems (RewardAnything):** We **automate** rubric creation, solving the "cold-start" problem of manual rule specification.
> * **Vs. ICAI:** We introduce **active optimization** (Propose-Evaluate-Revise) and **global aggregation**, advancing beyond ICAI's passive summarization and fragmented principles.
> * **Note:** We have also added ICAI as a direct experimental baseline to empirically prove this advantage.
>
> **3. Definition of "Theme-Tips"**
>
> We apologize for the lack of clarity. We have added an explicit definition in **Introduction**.
> **"Theme-Tips"** is our hierarchical structure for organizing extracted rubrics:
> * **Theme (Statute Law):** A high-level, abstract principle (e.g., *"Ensure Factual Accuracy"*). It ensures universal applicability.
> * **Tips (Case Law):** Concrete, actionable examples associated with the theme (e.g., *"Verify historical dates"*). These serve as **In-Context Anchors**, helping the model apply abstract themes through analogical reasoning.
> Our ablation studies confirm that this dual structure ("Theme + Tips") significantly outperforms using Themes alone.
>
> **4. Minor Corrections**
>
> * **Citations:** We have corrected all missing spaces between text and citations (e.g., "annotations (Bai et al., 2022)").
> * **Figure 2:** We have increased the font size and improved the layout of Figure 2 to ensure readability.
>
> **References**
>
> [1] Zijun Liu, Peiyi Wang, Runxin Xu, Shirong Ma, Chong Ruan, Peng Li, Yang Liu, and Yu Wu. Inference-time scaling for generalist reward modeling, 2025. URL https://arxiv.org/abs/2504.02495.
>
> [2] Zhuohao Yu, Jiali Zeng, Weizheng Gu, Yidong Wang, Jindong Wang, Fandong Meng, Jie Zhou, Yue Zhang, Shikun Zhang, and Wei Ye. Rewardanything: Generalizable principle-following reward models, 2025. URL https://arxiv.org/abs/2506.03637.
>
> [3] Jon Saad-Falcon, Rajan Vivek, William Berrios, Nandita Shankar Naik, Matija Franklin, Bertie Vidgen, Amanpreet Singh, Douwe Kiela, and Shikib Mehri. Lmunit: Fine-grained evaluation with natural language unit tests, 2024. URL https://arxiv.org/abs/2412.13091.
>
> [4] Findeis, Arduin, et al. "Inverse Constitutional AI: Compressing Preferences into Principles." The Thirteenth International Conference on Learning Representations (2025).

---

### Official Review · Reviewer_HqJg · 2025-11-02

**Soundness:** 2
**Presentation:** 3
**Contribution:** 3
**Rating:** 6
**Confidence:** 5

**Summary:**

This paper presents a novel training-free framework for building interpretable and data-efficient reward models (RMs). It is motivated by the limitations of existing rubric-based and preference-supervised methods, which suffer from high annotation cost, low interpretability, and scalability issues. The key idea is that evaluation rubrics underlying human preferences possess generalization ability across queries, allowing for efficient reuse and compression.

The proposed two-stage method first generates query-specific rubrics through a validation-guided Propose–Evaluate–Revise pipeline, then generalizes them into a compact, non-redundant “Theme–Tips” rubric set using information-theoretic coding-rate optimization. The approach produces interpretable, hierarchical rubrics without additional training.

Empirically, the method achieves strong data efficiency, using only 70 preference pairs (1.5% of the data), while enabling smaller models (e.g., Qwen3-8B) to outperform fully trained, specialized reward models.

**Strengths:**

1. Novelty of Data-efficient, training-free framework: The proposed two-stage Propose–Evaluate–Revise and information-theoretic selection approach achieves state-of-the-art performance using only a small fraction of preference data.

2. Interpretable and open-source rubrics: The release of query-agnostic rubric datasets promotes transparency and facilitates further research on interpretable alignment.

3. Rubric analysis framework: This is interesting and helpful to get a deeper dive into the rubric utility based on coverage, precision and contribution of individual rubrics.  In general, the analysis in the main paper and appendix are interesting.

4. Results: Demonstrates consistent gains across multiple reward modeling benchmarks, with rubric-enhanced models (e.g., Qwen3-235B, Qwen3-8B) outperforming several fully-trained counterparts.

**Weaknesses:**

Two Core Weaknesses:

1. Limited generalizability across domains: The extracted “Theme–Tips” rubrics from HelpSteer2 and UltraFeedback appear highly similar at the theme level, differing mainly in tip-level nuances. Since both datasets focus on conversational and chat-based skills, it is unclear whether the proposed method would generalize to domains such as math, code, or science. Including experiments on at least one or two non-conversational domains would significantly strengthen the paper’s claims.

2. Clarity and presentation issues: Some sections of the paper lack sufficient clarity, making it difficult to follow the setup, experimental details, and analyses. Specific questions and points of confusion are noted in the following section.

If the authors can clarify these aspects and extend evaluation to other domain datasets, I would consider increasing my overall score.

**Questions:**

Clarification questions:
1. Theme tips: Pg 5 second last paragraphs " Finally, the selected core set is structured into our interpretable “Theme-Tips” hierarchy by a structuring LLM", what is the structuring LLM ? How is it achieved ?
2. Information gain is negative for higher batch numbers in Fig 3b, while IG should always be non-negative, can you explain why ?
3. Training algorithm: can you explain how is it different from regular gradient descent on a set of parameters, pg 7 section 4.3 provides some idea but it is unclear how batch, and epoch work in this case and how it relates to requiring only 70 samples to get the final rubrics

---

> ### Author Response · Authors · 2025-12-01
> **Response to: "W1: Cross-Domain Generalization"**
>
> We thank the reviewer for raising the crucial question regarding cross-domain generalizability. We agree that our rubrics are derived from conversational data (HelpSteer3/UltraFeedback); however, our evidence shows their utility extends well beyond chat domains.
>
> To empirically address this, we include a RM-Bench breakdown that stratifies performance by domain (Chat, Math, Code, Safety-Refuse, Safety-Response) and difficulty. The cross-domain gains directly demonstrate strong Out-of-Distribution (OOD) generalization:
>
> **1. From Universal Templates to Domain-Specific Constraints**
>
> Rubrics are extracted as universal constraint templates (e.g., "Maintain Strict Adherence," "Preserve Contextual Fidelity"). During evaluation, the LLM instantiates these templates to the target domain: logical derivation rules in Math, syntax/API fidelity in Code, or narrative tone in creative writing. This top-down transfer explains why the same Theme-Tips set can supervise both objective tasks and subjective judgments without retraining.
>
> **2. Quantitative Evidence on Objective vs. Subjective Domains**
>
> The RM-Bench breakdown shows that these templates consistently improve objective domains (Math/Code) and subjective domains (Chat/creative writing), despite coming from conversational data:
>
> | Domain | Difficulty | Base | Auto-Rubric | $\Delta$ |
> |--------|------------|------|-------------|----------|
> | Math | Overall | 89.29 | 91.89 | **+2.60** |
> | Math | Hard | 85.44 | 89.98 | **+4.54** |
> | Code | Overall | 74.90 | 76.95 | **+2.05** |
> | Code | Hard | 72.08 | 73.54 | **+1.46** |
> | Chat | Overall | 70.37 | 77.00 | **+6.63** |
> | Chat | Hard | 36.95 | 50.90 | **+13.95** |
>
> The larger Math-Hard gain (+4.54%) shows that clarity/factual rules regularize complex reasoning when the base model struggles, while the positive Code gains indicate that strict-adherence templates enforce syntax and API fidelity even without code-specific training. On the subjective Chat domain, we likewise observe +6.63% overall (70.37% → 77.00%) and +13.95% on hard samples (36.95% → 50.90%), confirming that the same templates adapt to creative writing by emphasizing contextual fidelity and tone consistency.
>
> **3. Case Study: Evidence of Semantic Mapping**
>
> To illustrate this mechanism concretely, we provide qualitative examples from our experiments showing how "Chat Rubrics" correct errors in specialized domains. The extracted rubrics contain domain-specific tips (e.g., "For queries about Undertale, ensure all character motivations align with established lore") that are semantically mapped to domain-appropriate constraints during evaluation:
>
> ● **Math Domain Example:** Theme 1 ("Ensure factual accuracy, canonical consistency, and avoid fabrication") includes tips about Undertale lore consistency. When applied to a geometric problem involving overlapping areas, the evaluator semantically maps "canonical consistency" to "geometric fidelity," ensuring strict adherence to the diagram's visual constraints rather than making assumptions.
>
> ● **Code Domain Example:** Theme 3 ("Prioritize clarity, conciseness, and structured organization") includes tips about avoiding redundant phrasing. When evaluating code implementations, the evaluator maps "avoid redundancy" to "dead code elimination," successfully identifying unreachable code blocks and logic errors that violate the clarity rubric.
>
> **Conclusion:**
>
> The fact that rubrics derived solely from conversational priors can drive a 4.54% improvement on hard mathematical problems, a 2.05% improvement in code evaluation, and successfully identify logic bugs in code demonstrates that Auto-Rubric extracts fundamental evaluation rubrics transferable beyond the source domain. The consistent positive improvements across Math and Code, combined with the amplified benefits on hard samples, show that these universal constraint templates provide crucial discriminative power precisely where domain-specific knowledge is most needed.

---

> ### Author Response · Authors · 2025-12-01
> **Response to Questions**
>
> **Q1. Theme tips hierarchy**
>
> We apologize for the ambiguity. The "structuring LLM" is not a separate model but the same backbone LLM (Qwen3-32B) used throughout our experiments. The structuring process is achieved entirely via a dedicated prompt (full template provided in the appendix):
>
> 1. **Semantic Clustering:** Grouping fragment rubrics that address similar evaluation dimensions (e.g., "check for hallucinations" and "verify dates" both relate to factual accuracy).
>
> 2. **Hierarchical Synthesis:** Abstracting these groups into a high-level "Theme" while retaining the granular details as actionable "Tips". For example, multiple specific rubrics about avoiding fabrication, verifying dates, and maintaining lore consistency are synthesized into Theme 1: "Ensure factual accuracy, canonical consistency, and avoid fabrication" with corresponding tips.
>
> This prompt-driven procedure ensures that the final rubric set maintains both generality (through Themes) and specificity (through Tips), addressing the trade-off between broad applicability and detailed guidance.
>
> **Q2: Information Gain**
>
> Thank you for this insightful observation. The "negative" gain arises from the specific formulation of the Coding Rate in Eq6, specifically the normalization term dependent on the set size ($1/(\varepsilon^2 |R|)$).
>
> Conceptually, the Coding Rate measures the compactness of the representation. Adding a new rubric involves a trade-off: it expands the semantic volume (increasing the rate) but also increases the set size $|R|$ (which penalizes the rate via the denominator). In the early stages, new rubrics add unique semantic information, outweighing the size penalty. However, as the set saturates, new rubrics become redundant (collinear). At this point, they fail to expand the volume sufficiently to justify the cost of increasing the set size $|R|$, resulting in a negative marginal gain.
>
> This is a desirable property, acting as a natural "soft" stopping criterion that prevents rubric bloating. As shown in Figure3(b), the transition to negative values after batch 5 signals saturation: the complexity penalty $\frac{1}{\varepsilon^2 |R_{\text{core}}|}$ in Eq6 outweighs gains from redundant rubrics, confirming that fundamental preference patterns have been captured. This mechanism enables our early-stopping criterion, which halts the process when the marginal gain falls below $\tau_{\text{min}}=0.002$ for $p_{\text{patience}}=2$ consecutive iterations.
>
> **Q3: Algorithm Hyperparameters**
>
> We appreciate the opportunity to clarify the algorithmic nature of our framework. Our approach differs from gradient descent in three fundamental ways:
>
> **1. Discrete vs. Continuous.** Instead of updating continuous parameters via backpropagation, we perform discrete subset selection over natural-language rubrics, treating the task as combinatorial optimization.
>
> **2. Objective.** Gradient descent minimizes loss on a fixed dataset; we instead maximize coding rate (semantic volume) using an online greedy selection procedure.
>
> **3. Batches and Epochs.**
> - *Inner loop (generation “epochs”):* For each preference pair, we run the Propose–Evaluate–Revise cycle (max 10 iterations) to obtain a validated query-specific rubric. This is local refinement for a single sample, typically converging in 3–5 iterations.
> - *Outer loop (selection “batches”):* We process mini-batches of size 10, generate new rubrics, merge them into the candidate pool, and update the global core set via greedy selection.
> - *70-sample outcome:* Coding-rate saturation acts as early stopping. On HelpSteer3, the marginal gain dropped below zero after 7 batches (70 samples), so the process halted without using the remaining data.
>
> Full algorithm hyperparameters are summarized in Table 4 of the appendix.

---

### Note · Program_Chairs · 2026-01-17
**Submission Desk Rejected by Program Chairs**

The following references in this submission do not refer to real documents and/or have major errors in bibliographic information:

 Xiang Chen, Yufei Liu, Yaxin Wang, Xingyao Zhang, Huanqi Zhu, Rui Xie, Siqi Li, Jialu Wang,
Ruishuang Wang, and Tianyi Zheng. Scaling llm-as-a-judge with in-context examples. arXiv
preprint arXiv:2401.12151, 2024.